# Evidence of bifunctionality of carbons and metal atoms in catalyzed acetylene hydrochlorination

Vera Giulimondi[1], Andrea Ruiz-Ferrando[2,3], Georgios Giannakakis [1], Ivan Surin[1], Mikhail Agrachev[4], Gunnar Jeschke [4], Frank Krumeich [5], Núria López [2], Adam H. Clark [6] & Javier Pérez-Ramírez [1] ✉

Carbon supports are ubiquitous components of heterogeneous catalysts for acetylene hydrochlorination to vinyl chloride, from commercial mercury-based systems to more sustainable metal single-atom alternatives. Their potential co-catalytic role has long been postulated but never unequivocally demonstrated. Herein, we evidence the bifunctionality of carbons and metal sites in the acetylene hydrochlorination catalytic cycle. Combining *operando* X-ray absorption spectroscopy with other spectroscopic and kinetic analyses, we monitor the structure of single metal atoms (Pt, Au, Ru) and carbon supports (activated, non-activated, and nitrogen-doped) from catalyst synthesis, using various procedures, to operation at different conditions. Metal atoms exclusively activate hydrogen chloride, while metal-neighboring sites in the support bind acetylene. Resolving the coordination environment of working metal atoms guides theoretical simulations in proposing potential binding sites for acetylene in the support and a viable reaction profile. Expanding from single-atom to ensemble catalysis, these results reinforce the importance of optimizing both metal and support components to leverage the distinct functions of each for advancing catalyst design.

Carbons are common supports in metal-based heterogeneous catalysts[1,2], as exemplified in acetylene hydrochlorination to produce the vinyl chloride monomer (VCM, >40 Mton y$^{-1}$)[3,4]. The industrial technology utilizes toxic carbon-supported mercury(II) chloride catalysts[5,6]. 60 years of research efforts to identify sustainable metal alternatives have pinpointed precious metal chlorides, such as Au(I)Cl[6]. Recently, the isolated nature of these cationic Au species was identified by in situ X-ray absorption spectroscopy (XAS)[7]. This prompted the development of novel carbon-supported single-atom catalysts (SACs), defined as isolated active metal sites anchored to carbon functionalities[8,9]. Several precious metal SACs (e.g., Pt, Au, and Ru)

exhibit remarkable activities[6,10,11], yet Pt SACs stand out for their unparalleled stable performance[12]. Still, the catalytic behavior of any metal structure hinges on the use of carbons as supports, as other typical materials such as metal oxides result in limited activity[13–16]. Nevertheless, the underlying reason for this phenomenon remains unclear.

The importance of understanding the role of carbons cannot be overstated, as synthetic procedures to stabilize metal centers can inadvertently alter the carbon structure and surface chemistry[12,17], potentially compromising the efficacy of SACs[18]. In fact, carbons are known metal-free catalysts for acetylene hydrochlorination[16,19,20],

[1]Institute for Chemical and Bioengineering, Department of Chemistry and Applied Biosciences, ETH Zurich, Vladimir-Prelog-Weg 1, 8093 Zurich, Switzerland. [2]Institute of Chemical Research of Catalonia (ICIQ-CERCA), Av. Països Catalans 16, 43007 Tarragona, Spain. [3]Department of Physical and Inorganic Chemistry, Universitat Rovira i Virgili, Marcel·lí Domingo s/n, 43007 Tarragona, Spain. [4]Laboratory of Physical Chemistry, Department of Chemistry and Applied Biosciences, ETH Zurich, Vladimir-Prelog-Weg 1, 8093 Zurich, Switzerland. [5]Laboratory of Inorganic Chemistry, Department of Chemistry and Applied Biosciences, ETH Zurich, Vladimir-Prelog-Weg 1, 8093 Zurich, Switzerland. [6]Paul Scherrer Institute, 5232 Villigen, Switzerland. ✉e-mail: jpr@chem.ethz.ch

though the integration of metal components is essential to achieve the activity levels required for industrial competitiveness[6,16]. Furthermore, carbon surface functionalization treatments can effectively modulate the SAC activity, as demonstrated for commercial activated carbons and nitrogen-doped carbons[13,17]. Nevertheless, due to a lack of conclusive experimental evidence, the potential contribution of carbon in fulfilling the catalytic cycle has only been postulated[21]. Limitations of conventional characterization tools in assessing active site structures have led to investigations relying mainly on theoretical simulations. Still, owing to the complexity and heterogeneity of carbon surfaces, theoretical studies focus solely on metal atoms when considering active sites[3,22]. A recent study exploring the potential contribution of carbons suggested that surface functionalities activate hydrogen chloride[23]. In addition, carbon porosity and surface functionalization were experimentally shown to regulate acetylene adsorption[16], though the acetylene activation step was attributed to the metal sites based on computations. To determine the exact function of each catalyst component, key to deriving accurate synthesis-structure-performance relations, advanced characterization techniques constitute valuable tools. By providing time-resolved and environment-sensitive information, inaccessible from ex situ measurements, in situ and *operando* studies, can establish the nature of active sites and track their dynamic behavior[24–29]. The accuracy can be enhanced by employing recently-developed data analysis tools, providing insights into the kinetics of active site structural evolution[28,29]. Thereby, the catalytic function of carbons in acetylene hydrochlorination could be unequivocally identified.

Herein, we provide evidence for the active participation of carbons in the catalytic cycle of acetylene hydrochlorination, working in tandem with metal centers. We explore a platform of single metal atoms (Pt, Au, Ru) supported on commercial activated carbon, a non-activated counterpart, and a nitrogen-doped carbon, exhibiting distinct performances. Combining *operando* X-ray absorption spectroscopy with electron paramagnetic resonance and X-ray photoelectron spectroscopic analyses as well as kinetic investigations, we give insights into the chemical state of metal and support sites during synthesis, as well as their individual interactions with reactants under acetylene hydrochlorination. Expanding current definitions in single-atom catalysis, we identify that active sites comprise metal atoms, activating hydrogen chloride, and metal-neighboring sites in the support, binding acetylene. Guided by *operando* techniques resolving the metal coordination environment under reaction conditions, theoretical simulations enable us to propose potential binding sites for acetylene in the support and a viable reaction profile over bifunctional ensembles formed by metal-carbon sites. Our results highlight the importance of selectively engineering the metal and carbon sites as a promising strategy to maximize their distinct catalytic function and unlock superior performance.

## Results

### Platform of single-atom catalysts and structural evolution during synthesis

To systematically assess the respective role of the metal and carbon components in acetylene hydrochlorination, we generate a platform of Pt, Au, and Ru single atoms (metal content, 1 wt%) supported on a commercial activated carbon (AC), provided the industrial application of ACs in the technology[6]. Furthermore, by taking advantage of the intrinsic stability of Pt atoms on carbon supports[16], Pt SACs are synthesized on a commercial non-activated AC analog (C) and a nitrogen-doped carbon (NC) to compare the effect of different porous properties and surface functionalities, respectively (Supplementary Table S1). To gain insights into synthesis-structure-performance relations, we probe the structural evolution of the metal and carbon components from catalyst synthesis to operation (Supplementary Tables S2). For this purpose, we employ *operando* XAS to monitor the

dynamic behavior of metal atoms, i.e., evolution of the metal oxidation and coordination states. Owing to the highly corrosive nature of the reaction environment, requiring careful experiment design and safety assessment (see Supplementary Information), very few in situ and *operando* XAS studies have been conducted to date[3,11]. These exclusively focused on the metal sites. In this study, we consider both the metal and carbon components, complementing XAS measurements with X-ray photoelectron spectroscopy (XPS) to assess changes in the carbon structure. Furthermore, the formation of carbonaceous deposits is analyzed by electron paramagnetic resonance spectroscopy (EPR, Supplementary Table S3). Finally, guided by the resolution of the working metal atom coordination through *operando* studies, theoretical simulations are employed to explore the nature of the reactant binding sites and propose a reaction profile (Fig. 1).

The catalysts are synthesized by incipient wetness impregnation (IWI) of the selected carbon with a solution of the desired metal chloride precursor (e.g., $H_2PtCl_6$), followed by drying and thermal activation (Fig. 2a, Supplementary Table S4). The obtained catalysts are denoted as $M_X$/support-solvent-$T_a$ (M = Pt, Au or Ru; $X$ = single atoms (SA) or nanoparticles (NP), support = AC, NC, or C; solvent = water (w) or aqua regia (a); $T_a$ = 473–1073 K; example: $Pt_{SA}$/AC-w-473, Supplementary Table S1). High metal dispersion is corroborated by X-ray diffraction (XRD, Supplementary Fig. S1) and high-angle dark-field scanning transmission electron microscopy (HAADF-STEM, Supplementary Fig. S2). To monitor the process of metal atom stabilization over the support, the thermal activation of Pt species is investigated by *operando* XAS over oxygen functionalities, in AC, and nitrogen ones, in NC, as they exhibit different affinity for the metal species (Supplementary Figs. S3, S4, Supplementary Tables S5, S6). Upon thermal treatment, the chlorinated Pt species undergo a gradual loss in chloride ligands, as evidenced by a drop in the white line intensity of the X-ray absorption near edge spectroscopy (XANES) spectrum (indicating a shift from a $H_2PtCl_6$-like to a $PtCl_2$-like state, Fig. 2b, Supplementary Fig. S4). This is followed by a progressive anchoring process over the support, reflected by a shift in the white line position and a slight increase in its intensity, starting to resemble the $PtO_2$ reference (Fig. 2b, Supplementary Fig. S4). The thermal treatment effect on the metal speciation depends on the strength of the metal interaction with coordinating functionalities on the support[12]. In line with previous reports[12], mild activation temperatures (i.e., 473 K) yield single atoms stabilized by chloride ligands. Harsher thermal treatments (i.e., > 673 K) lead to chloride-ligand removal[12], resulting in metal sintering on AC already at 873 K, and the formation of chloride-free isolated Pt sites on NC that are four-fold coordinated with the support at 1023 K (Fig. 2c), as corroborated by extended X-ray absorption fine structure (EXAFS, Supplementary Table S6). The multivariate curve resolution (MCR) analysis is a valuable tool to process a large number of XAS spectra, and accurately resolve changes over time[28]. The evolution of the spectral features of the Pt species can be described by three components as determined by singular value decomposition analysis (Fig. 2d, Supplementary Fig. S5). These can be individually compared with the $H_2PtCl_6$, $PtCl_2$, and $PtO_2$ references, respectively reflecting the evolving "chlorinated", "partially dechlorinated", and "support-anchored" metal speciation during the thermal treatment (Supplementary Fig. S6). Furthermore, the MCR analysis shows that the kinetics of partial dechlorination and anchoring on the support of Pt species is slower on AC compared with NC, where the thermal activation process is fully accomplished already during the temperature ramp (Fig. 2d). Reflecting the firmer stabilization of the metal sites on NC compared with AC, these results experimentally reaffirm the computationally-derived insights on the different affinity of metal species for oxygen- and nitrogen-functionalities in carbon hosts[12].

Besides thermal activation, another parameter affecting the metal speciation is the solvent employed to dissolve the metal precursor in

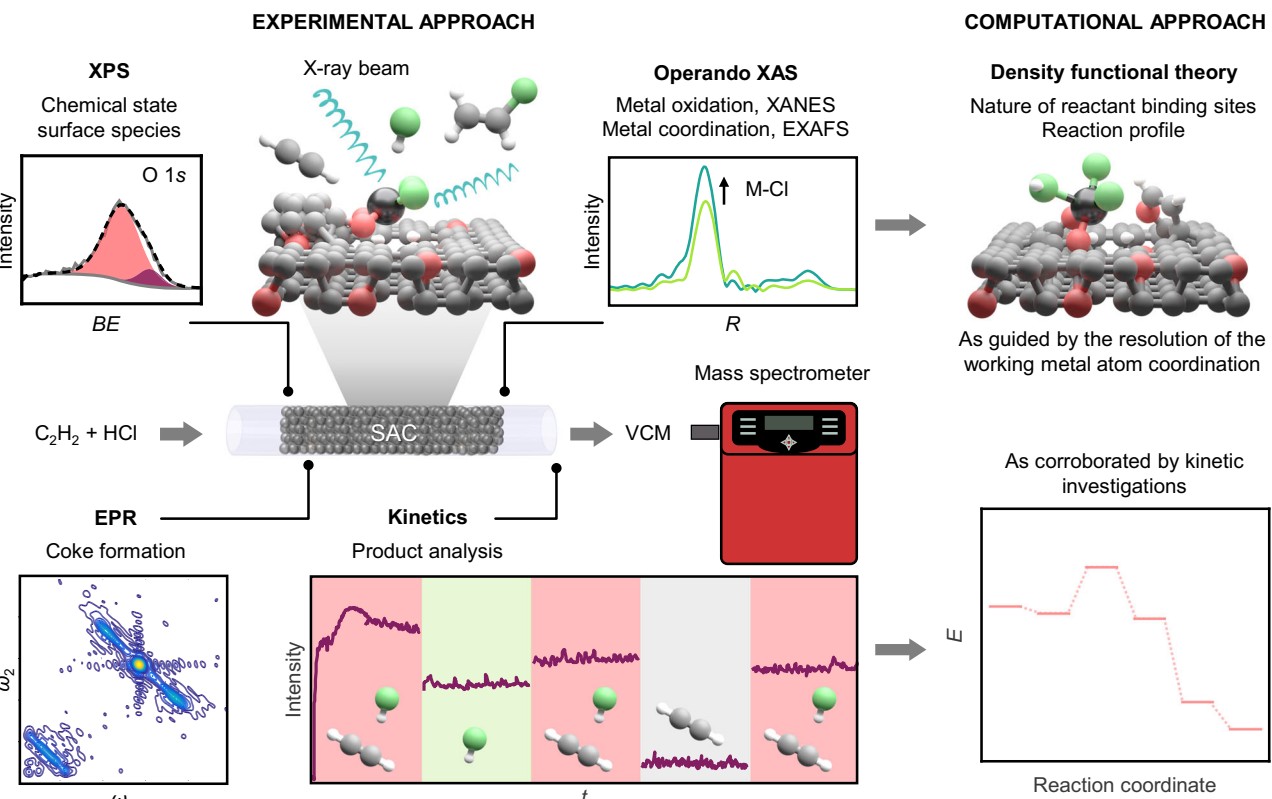

**Fig. 1 | Approach to resolving the catalytic role of carbon supports.** Schematic of the multi-technique strategy employed to resolve the role of carbon and metal sites in catalyzing acetylene hydrochlorination. The chemical state of surface species, encompassing carbon functionalities, chlorine, and metal atoms is evaluated by XPS. *Operando* XAS enables monitoring of the oxidation (by XANES) and coordination (by EXAFS) states of working metal sites. Combination with EPR analysis of carbonaceous species and kinetic investigations permits to gain insights into the catalytic function of carbons. Integratively with experimental investigations, theoretical simulations shed further light on the nature of the reactant binding sites and propose a reaction profile.

the IWI step[17,30]. Specifically, carbons present surface functionalities that can reduce the deposited metal species. The use of highly oxidizing agents, such as aqua regia, can prevent this phenomenon and yield oxidized chloride-stabilized single metal atoms. This is exemplified by the Au and Ru SACs, requiring the use of such solvents to avoid metal sintering under thermal activation[30]. Conversely, the tendency of Pt species to stabilize as single atoms on carbon surfaces, regardless of the employed solvent, enables us to conduct a comparative analysis of the properties of aqua regia- and water-derived SACs. The oxidizing solvent can have a dual-level impact on the SAC structure: the generation of metal atoms with higher chlorination degree and the modification of the carbon functionalities. While the latter is acknowledged in the literature[7], its effect has been primarily studied in terms of dispersion and nature of the metal species. Comparison of the initial catalytic activity of the water- and aqua regia-derived Pt SACs shows a noticeable drop for the latter case when supported on both AC and NC (Fig. 3a). This might be ascribed to the higher chlorination degree of the aqua regia-derived Pt single atoms, hindering the metal interaction with the reactants[31]. However, temperature-programmed desorption of acetylene coupled to mass spectrometry ($C_2H_2$-TPD-MS) shows loss in the acetylene adsorption capacity, regulated by the carbon[16], upon impregnation with aqua regia as opposed to water (Fig. 3b). This can be attributed to higher chlorination of the support and alterations of surface oxygen functionalities caused by the oxidizing solvent, as corroborated by XPS (Fig. 3c, d, Supplementary Fig. S7, Supplementary Tables S7–S10). To explore the effect of carbon surface alteration in adsorbing acetylene, the bare support was impregnated with a metal-free aqua regia solution, dried at 473 K, subsequently impregnated with a $H_2PtCl_6$-containing aqueous solution, and finally thermally activated at 473 K. The resulting catalyst presents comparable catalytic

activity and acetylene adsorption properties to those of the aqua-regia-derived analog, $Pt_{SA}$/AC-a-473, which are significantly reduced compared with those of the water-derived counterpart, $Pt_{SA}$/AC-w-473 (Supplementary Fig. S8). Furthermore, the similar acetylene adsorption properties exhibited by the bare carbon support and $Pt_{SA}$/AC-w-473 suggest carbon functionalities as acetylene-binding sites rather than the chlorinated Pt atoms (Supplementary Fig. S8).

**Dynamic behavior of metal sites under reactive environments**
To investigate the role of metal atoms and carbon supports in catalyzing acetylene hydrochlorination, the metal coordination environment in $Pt_{SA}$/AC-w-473 under reaction conditions is monitored by *operando* XAS (Fig. 4a, b, Supplementary Fig. S9, Supplementary Tables S5, S11). Although the catalyst undergoes initial moderate loss in activity (−27% over 10 h on stream), virtually no changes in the metal state are detected, pointing to modifications in the carbon (e.g., coke formation blocking reactant access to the metal sites) as the underlying cause for deactivation rather than metal-related mechanisms previously postulated, such as overchlorination or changes in the oxidation state[12,31]. Detailed analysis of the EXAFS spectra shows a slight increase in the metal site chlorination (coordination number, CN = 2.4 to 2.7, Supplementary Table S11) under reaction conditions, indicating hydrogen chloride activation. Notably, the Pt-C/O contributions, deriving from the metal coordination with the support, remain unaltered upon feeding the reaction mixture (Supplementary Table S11), pointing to the absence of metal-acetylene interactions. Previously simulated reaction mechanisms proposed a catalytic cycle involving acetylene adsorption over the metal sites and subsequent reaction with gas-phase hydrogen chloride to form VCM[3,12]. In contrast, our *operando* analysis suggests that the acetylene activation step takes

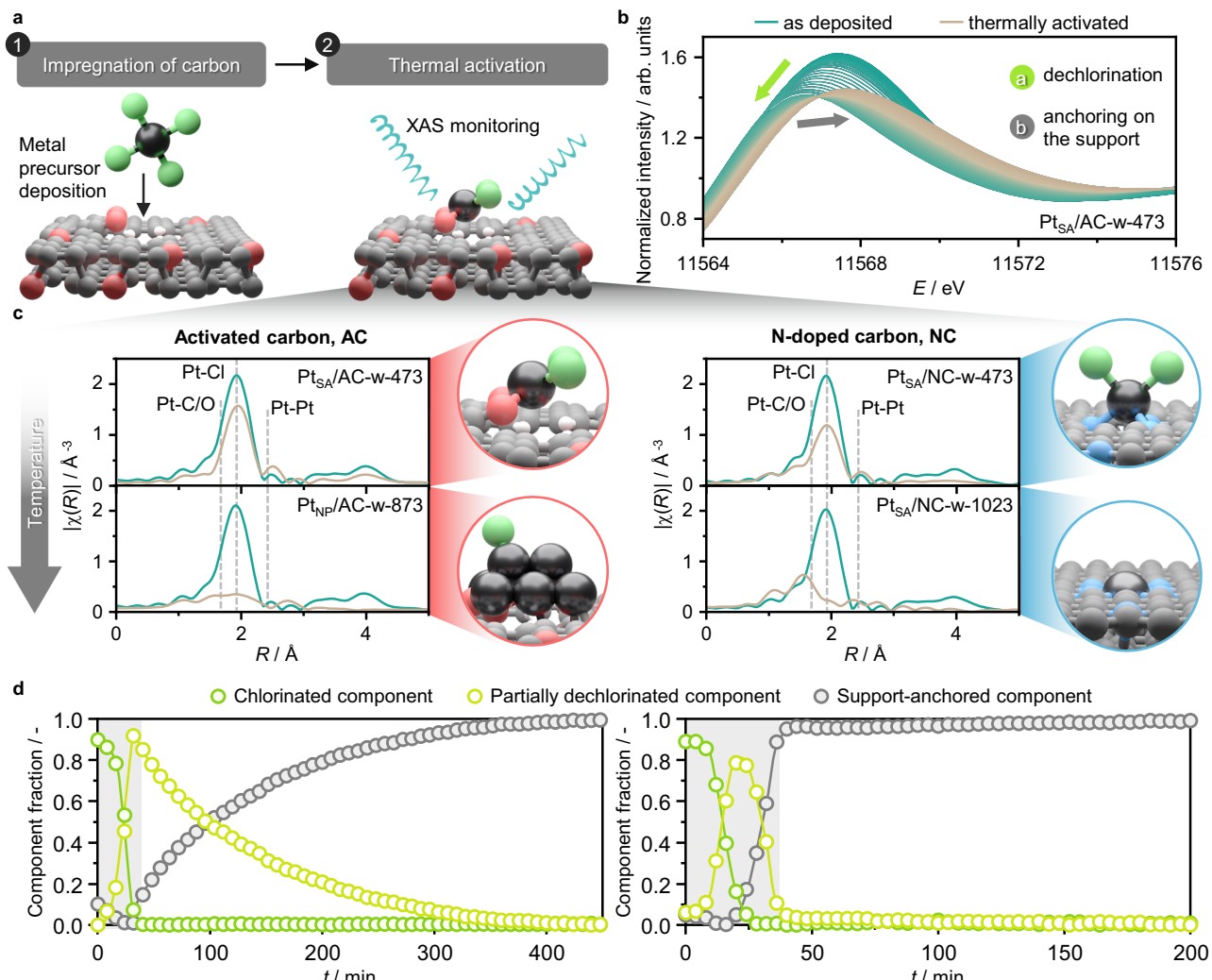

**Fig. 2 | Synthesis and *operando* XAS characterization of Pt SACs. a** Schematic representation of the synthetic strategy employed. The thermal activation step is monitored by XAS, consisting of a temperature ramp (5 K min⁻¹) and a thermal dwell (473 or 1073 K). **b** *Operando* Pt $L_3$ edge XANES of $H_2PtCl_6$ deposited on AC in an aqueous solution under thermal activation (473 K). **c** *Operando* Pt $L_3$ edge EXAFS of $H_2PtCl_6$ deposited on AC and NC in an aqueous solution under thermal activation (473 and 1073 K). Owing to the sensitivity of EXAFS to thermal effects, preventing comparison of spectra acquired during the temperature ramp, the as deposited (green) and thermally activated (brown) states are shown, collected at room temperature. **d** MCR analysis of *operando* XAS of $H_2PtCl_6$ deposited on AC (left) and NC (right) in an aqueous solution during the temperature ramp (gray region) and thermal dwell (473 K, white region).

place over the carbon support, while the metal sites bind hydrogen chloride.

Aiming to evaluate the nature and strength of the metal interaction with each reactant, AC-supported Pt, Au, and Ru SACs are exposed to a sequence of reactive environments (Fig. 4c) at 473 K. These include (i) helium, providing an inert reference state, (ii) the reaction mixture, to study the reactant preferential adsorption on the metal sites, (iii) hydrogen chloride, (iv) the reaction mixture, to explore if the metal sites can undergo deactivation by overchlorination, (v) acetylene, to probe whether the reactant can adsorb over the metal atoms in the absence of hydrogen chloride, and (vi) the reaction mixture, to investigate if exposure to acetylene alters the catalytic properties of the metal sites. Consistently with previous observations (Fig. 4a), slight metal chlorination is observed in the Pt SACs under the reaction mixture and no interaction with acetylene is detected, while exposure to only hydrogen chloride leaves the metal sites virtually unaltered (Fig. 4d, e, Supplementary Fig. S10, Supplementary Tables S5, S12). On the contrary, upon exposure to only acetylene, chloride ligands are partially removed. The subsequent exposure to the reaction mixture does not restore the initial chlorination degree (CN = 1.6 *vs.* 2.5), while

the metal-carbon interactions remain unaltered. This could be ascribed to changes in the metal oxidation state or coke formation on the support, blocking reactant (i.e., hydrogen chloride) access to the metal sites. Interestingly, product analysis by mass spectrometry shows that VCM evolution is still detected upon removal of acetylene from the reactant stream, consistent with the proposed role of carbon as an "acetylene reservoir"[16], and extinguished when hydrogen chloride is no longer fed (Fig. 4f). Insights into the interaction of acetylene with the carbon support is gained through analysis of the chlorine and oxygen surface species by XPS, after exposure to the relevant reactive environments (Supplementary Fig. S11, Supplementary Tables S9, S10). In line with the EXAFS results, analysis of the Cl $2p$ XPS spectra after exposure to acetylene only (reactive environment *v*, Fig. 4c) reveals a prominent reduction in the chlorine-metal contribution, corroborating the chloride-supplying role of the metal in the catalytic cycle. Conversely, the chlorine-carbon contribution is preserved, suggesting that the support binds chloride ions too strongly to catalyze VCM evolution. Furthermore, analysis of the O $1s$ spectra shows that the oxygen functionalities undergo a slight reduction, pointing toward their participation in the reductive formation of VCM. Analysis of the

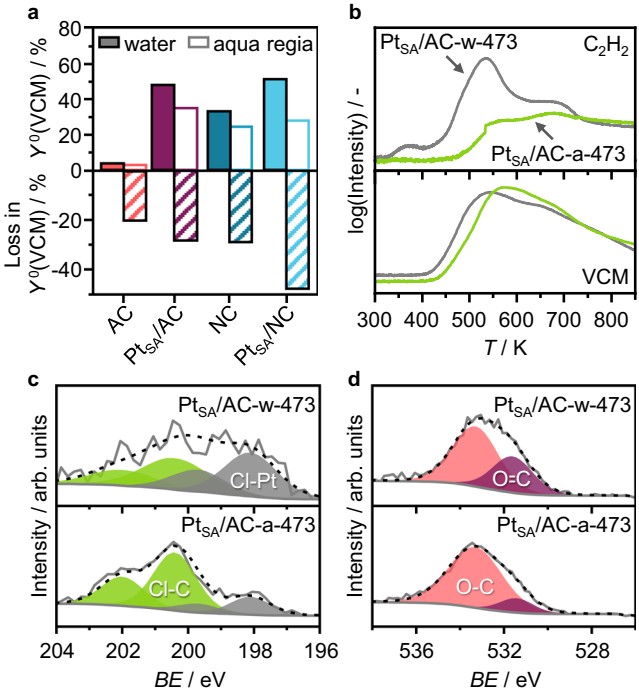

**Fig. 3 | Impact of carbon support modifications during synthesis. a** Initial activity (expressed as VCM yield after 1 h on stream, $Y^0$(VCM), top) of the Pt SACs, derived from carbon impregnation with $H_2PtCl_6$ in aqueous and aqua regia solutions, and the bare carbons, impregnated with the same solvent, together with the loss in activity (striped bars, bottom) suffered from catalysts exposed to aqua regia compared with water. Reaction conditions: $T_{bed} = 473$ K, $F_{tot} = 15$ cm$^3$ min$^{-1}$, $C_2H_2$:HCl:Ar = 40:44:16, $W_{cat} = 0.25$ g. **b** Time-resolved product analysis over AC-supported Pt SACs in $C_2H_2$-TPD-MS analysis. **c** Cl 2$p$ and **d** O 1$s$ XPS spectra of AC-supported Pt SACs.

dynamic behavior of AC-supported Au and Ru SACs corroborates the bifunctional role of metal atoms and carbons (Fig. 4e). Although exhibiting distinct features, as Au atoms undergo metal sintering while Ru atoms are prone to further chlorination, neither Au nor Ru atoms show coordination with acetylene (Supplementary Figs. S12, S13, Supplementary Tables S13, S14).

**Carbon properties and metal ligand effects**

To compare the effect of different surface functionalities and porous properties on the metal site behavior, we monitor Pt SACs supported on NC and C under exposure to the sequence of reactive environments shown in Fig. 4c (Fig. 5a, Supplementary Figs. S14, S15, Supplementary Tables S5, S15, S16). Exhibiting high surface area, NC yields a Pt SAC with comparable initial activity as AC, which, however, undergoes fast and pronounced deactivation (Fig. 5b)[12]. Despite the distinct performance over time, the metal sites show a similar dynamic behavior to their AC-supported counterparts (Figs. 4e, 5a), exhibiting no coordination with acetylene. In line with the computationally-predicted tendency of pyrrolic N-sites to promote coke formation[12], this suggests acetylene polymerization over N-functionalities as the leading cause for deactivation. The importance of the carbon support for achieving high performance is exemplified by the Pt SAC supported on the commercial non-activated carbon, Pt$_{SA}$/C-w-473. The two-order-of-magnitude lower porosity of this support, compared with the AC-analog (Supplementary Table S4), leads to a drastically reduced catalytic activity (Fig. 5c). The limited porosity is linked to a lack of acetylene adsorption capacity[16], preventing the carbon from acting as an "acetylene reservoir". The different properties of the carbon support also result in unusual dynamic behavior of the metal atoms under reactive environments. Though no coordination with acetylene is

observed, similarly to their AC- and NC-supported counterparts, the Pt species are prone to agglomerating (Fig. 5a, Supplementary Table S16). This tendency is ascribed to the lesser stabilization provided by the non-activated support, featuring lower surface area and distinct functionalization[16]. Unaltered under reaction conditions, the metal atoms undergo a sintering process that is triggered upon sole exposure to hydrogen chloride and maintained thereafter. This might be attributed to higher chlorination of the Pt atoms under hydrogen chloride, resulting in mobile metal species[32]. In a dynamic process, these can form metal-metal bonds via the reductive elimination of hydrogen chloride or chlorine.

The lack of metal-acetylene interactions in all the examined Pt SACs supported on the diverse carbons (AC, NC, and C), indicates that the acetylene activation step occurs over the carbon, irrespective of its properties. The acetylene adsorption on the metal sites might be prevented by the presence of chloride ligands[31]. Computational investigations on their role in determining the activity of the metal atoms suggested that highly chlorinated metal single atoms suffer from low acetylene affinity, identifying metal chlorination on stream as a deactivation mechanism[12,31]. As the entire catalytic cycle was assumed to be fulfilled over the metal sites, the adsorption and subsequent activation of hydrogen chloride on the metal site as the first reaction step was identified as less energetically favorable than acetylene activation, leading to closing the Pt coordination sphere and hence compromising the acetylene affinity. This could be experimentally investigated by monitoring aqua regia- and water-derived AC-supported Pt SACs, featuring different metal chlorination degrees, under reactive environments (Fig. 4c). However, the two Pt SACs exhibit a similar metal atom dynamic behavior under reactive environments (Figs. 4e, 5a, Supplementary Fig. S16, Supplementary Table S17). The highly chlorinated aqua regia-derived Pt single atoms appear to undergo slight dechlorination over time on stream attributable to VCM formation, likely occurring on the carbon. The loss in chloride ligands is maximized under exposure to only acetylene. Accordingly, the Pt single atoms reach the same chlorination degree as their water-derived counterparts (Fig. 4d). To probe metal-acetylene interactions in the absence of chloride ligands, AC-supported Pt single atoms derived from an aqueous solution of a chlorine-free platinum potassium cyanide precursor, Pt$_{SA}$(CN)/AC-w-473, are probed under reaction conditions by *operando* XAS (Supplementary Fig. S17, Supplementary Table S18). Rapid chlorination is observed, indicating hydrogen chloride adsorption and activation, while no metal-acetylene coordination is detected (Fig. 5d). The progressive chlorination of the metal sites reflects in increasing catalytic activity over time (+18% over 12 h on stream, Fig. 5e, f), indicating that metal chlorination on stream does not constitute a deactivation mechanism.

**Catalytic role of metal-neighboring sites in the carbon support**

To gain deeper insights into the dynamic behavior of metal sites in the presence of acetylene, Pt SACs, supported on different carbons and prepared with different synthetic procedures, are continuously exposed to the reactant during a temperature ramp from room temperature to reaction temperature, i.e., 473 K (Fig. 6a). While partial metal dechlorination is observed as a common feature across the diverse samples, the extent of the phenomenon depends on the carbon properties and synthetic procedure (Fig. 6b, Supplementary Fig. S18, Supplementary Tables S5, S19, S20). For example, the chlorinated Pt species are found more stable on NC than AC, in line with previous results (Fig. 2c), and undergo smaller loss of chloride ligands. The stability of the N-coordinated Pt sites is maximized when activated at high temperature: exhibiting full coordination with the support, they undergo virtually no structural changes under acetylene. Their square-planar geometry results in reduced activity, which is comparable to the one of the bare NC support and attributable to the low affinity of the metal atoms for hydrogen chloride

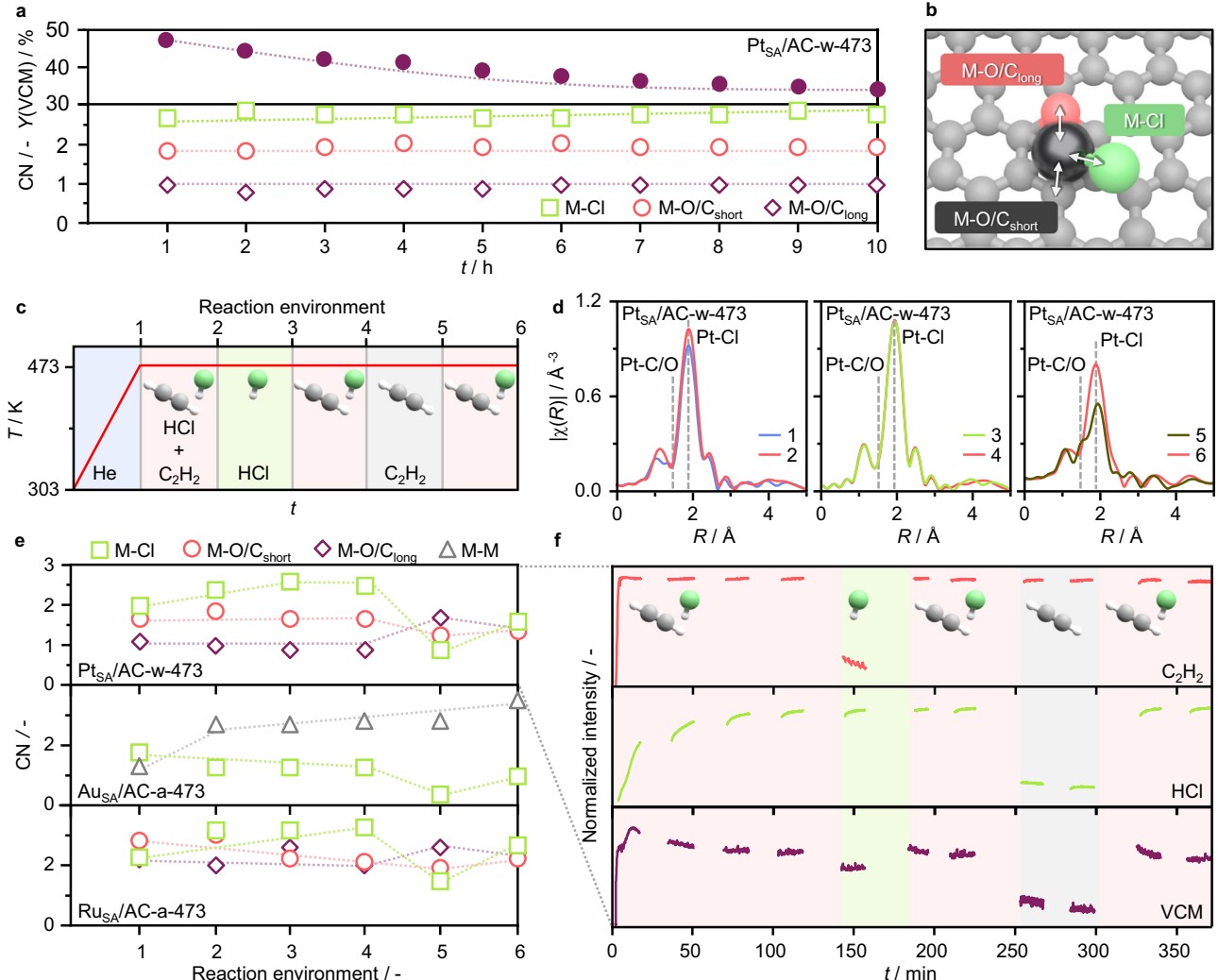

**Fig. 4 | Resolution of the catalytic roles of carbon supports and metal sites.**
**a** Time-on-stream performance of the catalyst (top) and *operando* Pt $L_3$ edge
EXAFS-derived coordination numbers (CN, bottom) of $Pt_{SA}$/AC-w-473. Reaction
conditions: $T_{bed}$ = 473 K, $F_{tot}$ = 15 cm³ min⁻¹, C₂H₂:HCl:Ar = 40:44:16, $W_{cat}$ = 0.25 g.
**b** Schematic representation of metal-neighboring scattering atoms contributing to
the XAS spectrum. **c** Sequence of reactive environments to which metal sites are
exposed during XAS monitoring. **d** *Operando* Pt $L_3$ edge EXAFS of $Pt_{SA}$/AC-w-473.
**e** *Operando* Pt $L_3$, Au $L_3$ and Ru $K$ EXAFS-derived CN for $Pt_{SA}$/AC-w-473, $Au_{SA}$/AC-a-
473, and $Ru_{SA}$/AC-a-473. **f** Time-resolved product analysis by mass spectrometry
over $Pt_{SA}$/AC-w-473 as a function of time-on-stream.

(Fig. 5c)[12]. Conversely, the prominent dechlorination undergone by the
AC-supported Pt SACs reflects in greater metal-C/O contributions. At
first sight, these might hint at the formation of metal-acetylene bonds.
Nevertheless, EXAFS analysis evidences two contributions: an unal-
tered short-bonding one, at 1.85 ± 0.03 Å, and a growing long-bonding
one, at 2.04 ± 0.02 Å (Supplementary Table S19). Though oxygen and
carbon atoms exhibit similar scattering properties in XAS, the short-
and long-bonding contributions are in agreement with Pt-C and Pt-O
references, respectively (Fig. 4b)[33,34]. Consequently, although some
degree of metal-acetylene contributions cannot be ruled out, the
increase in the Pt-O contribution likely derives from the stronger
interaction of the metal sites with the support upon dechlorination.
The relation between the structural stability of the chlorinated Pt
species and the dechlorination extent is corroborated by the greater
loss in chloride ligands exhibited by the aqua regia-derived AC-sup-
ported Pt SAC compared to its water-derived analog. The oxidizing
solvent yields highly chlorinated Pt species that could be, conse-
quently, more prone to donating more ligands to the acetylene mole-
cules for VCM formation[12]. Indeed, detailed spectral analysis by
MCR, which can be described by two components as determined by
singular value decomposition analysis (Supplementary Fig. S19), shows

that the dechlorination process is triggered during the temperature
ramp, at approximately 373 K (Fig. 6c, Supplementary Fig. S20). As a
result, the phenomenon is attributed to the reaction of the chloride
ligands of the metal atoms with acetylene molecules adsorbed on
neighboring sites in the carbon carrier to form VCM. This is corrobo-
rated by C₂H₂-TPD-MS analysis of $Pt_{SA}$/AC-w-473 and $Pt_{SA}$/AC-a-473,
forming similar amounts of VCM even though the acetylene adsorp-
tion capacity is reduced in the latter (Fig. 3b). This suggests that only
acetylene molecules adsorbed in the vicinity of metal sites react to
generate VCM, and its amount is dictated by the available chlorides
supplied by the metal sites.

The participation of metal-neighboring sites of the carbon sup-
port in the catalytic cycle is further validated by analysis of the car-
bonaceous species formed under reaction conditions by EPR. While
the continuous wave EPR spectrum of as-prepared $Pt_{SA}$/AC-w-473
shows no signals, the spectrum of the same catalyst exposed to the
reaction mixture for 12 h exhibits a sharp symmetric line with
approximately Lorentzian lineshape, centered at $g$ = 2.003 (Supple-
mentary Fig. S21). Such signal is typical for organic radicals (S = 1/2) and
can be attributed to radical compounds generated during the reaction,
forming coke[35]. While the Lorentzian lineshape reflects high local

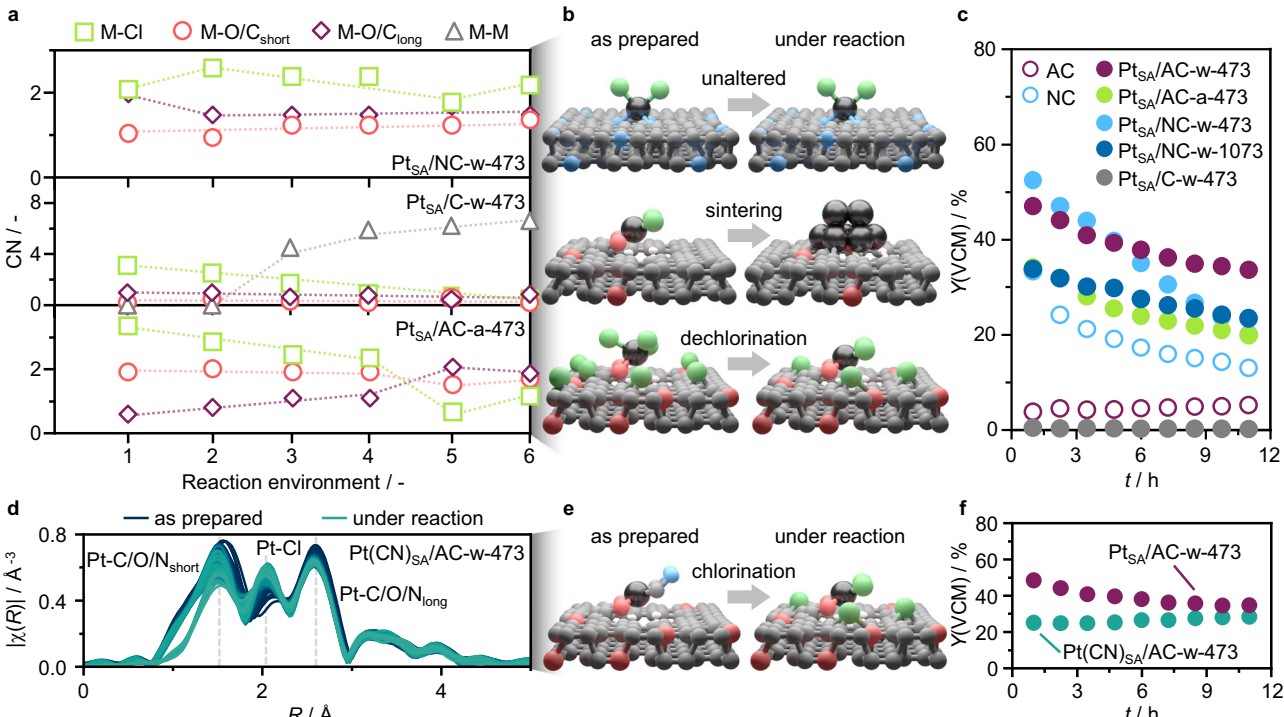

**Fig. 5 | Carbon property and metal ligand effects. a** *Operando* Pt $L_3$ edge EXAFS-derived CN for Pt SACs, exposed to reaction environments shown in Fig. 3c, supported on NC, C, and AC, yet derived from impregnation with $H_2PtCl_6$ in an aqua regia solution, accompanied by **b** structural representation of the metal atom dynamic behavior under reaction conditions. **c** Time-on-stream performance of the Pt SACs, and the bare AC and NC supports, provided as references. **d** *Operando* Pt $L_3$ edge EXAFS of Pt(CN)$_{SA}$/AC-w-473 under acetylene hydrochlorination conditions at 473 K, accompanied by the **e** structural representation of the metal atom dynamic behavior under reaction conditions, and the **f** Time-on-stream performance. Reaction conditions: $T_{bed}$ = 473 K, $F_{tot}$ = 15 cm$^3$ min$^{-1}$, $C_2H_2$:HCl:Ar = 40:44:16, $W_{cat}$ = 0.25 g.

coke density, the low intensity of the signal shows that the average concentration is very low. This suggests that coke is mostly localized in proximity to the active sites and generated from the polymerization of reaction intermediates. To verify this, we employ the Hyperfine Sublevel Correlation Spectroscopy (HYSCORE), a pulsed EPR technique that can detect weak hyperfine couplings between coke radicals and nearby magnetic nuclei (Fig. 6d, Supplementary Fig. S22). The HYSCORE spectrum shows strong couplings with $^1H$ (I = 1/2) and $^{13}C$ (I = 1/2) nuclei of the polyaromatics and a weaker signal resulting from their interaction with $^{195}Pt$ (I = 1/2, 34% isotopic abundance). Analysis of the latter (see Supplementary Information) enables us to determine the spin density in the Pt s-orbitals ($\approx 7.6 \times 10^{-5}$), and evaluate the interaction between coke radicals and Pt sites assuming negligible and pure through-space coupling (i.e., dipole-dipole interactions). In the former case, we estimate the spin density in the Pt d-orbitals to be up to $4.7 \times 10^{-4}$. Such low value suggests the absence of direct coke-Pt bonds, since close proximity between coke and metal sites would reflect in high spin density. Consistently, assuming pure through-space coupling, we assess an average coke-Pt distance of about 5 Å. These results further corroborate that acetylene adsorbs and reacts on the support, forming coke deposits in proximity of Pt atoms, rather than directly on them.

**Fulfillment of the catalytic cycle**

*Operando* XAS analyses indicate that metal atoms activate hydrogen chloride, while $C_2H_2$-TPD-MS and EPR investigations evidence the ability of the carbon support to bind acetylene. Thereafter, integrative kinetic and computational investigations are conducted to probe whether both reactants partake in the catalytic cycle in their adsorbed state, and the nature of the binding site for each reactant. These efforts are focused on Pt$_{SA}$/AC-w-473, selected for its high

activity and unparalleled stability[12]. First, partial reaction orders of acetylene and hydrogen chloride were derived. The obtained values (0.60 and 0.48, respectively, Supplementary Fig. S23), lower than 1, suggest that none of the reactants participates in the catalytic cycle in the gas-phase[36]. This observation is in line with previous kinetic studies on Au SACs by temporal analysis of products, pointing to the adsorption of both reactants on the catalyst surface as essential for optimal catalytic performance[37]. On the basis of these observations, density functional theory (DFT) simulations proposed a viable reaction profile wherein both reactants are adsorbed and activated over carbon-supported monochlorinated Au and Pt metal atoms[12,37]. Indeed, in the absence of *operando* characterization, single-atom catalysts and reaction mechanisms are often modeled according to organometallic chemistry principles owing to the structural similarities between SACs and organometallic catalysts[38], centered on the ability of the metal atoms to coordinate with reactants. Nevertheless, resolving experimentally the coordination environment of the metal atoms under reaction conditions can enable a more accurate identification of their catalytic role by theoretical methods.

Herein, guided by *operando* XAS analyses, DFT simulations are conducted to probe the nature of the binding sites for each reactant. Diverse active structures featuring Pt atoms with varying chlorination degrees and stabilized on different oxygen-containing anchoring sites in the carbon support are considered. Epoxide, keto, hydroxyl, and carboxylic acid groups in different lattice configurations, namely epo$_2$, OH, keto$_2$, keto$_4$, edge-CO, and edge-COOH, are selected to represent the possible chemical environments in AC[31,39]. These structures are evaluated on their ability to stabilize PtCl$_x$ (x = 0−3) species (Supplementary Table S21). In line with the *operando* XAS analysis, indicating that the Pt atoms are three-fold coordinated with the support (Pt-C/O CN = 2.7, Supplementary Table S11) and presenting at least two

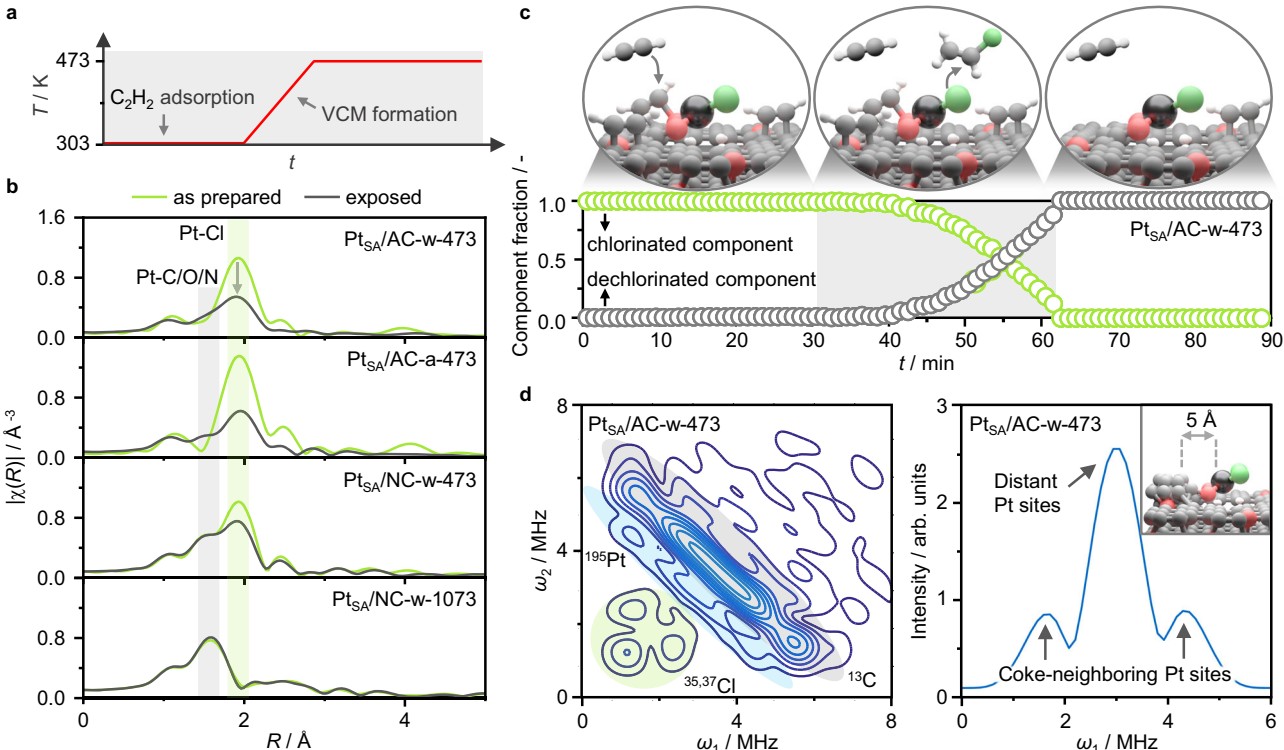

**Fig. 6 | Acetylene interaction with carbon support and metal sites. a** Schematic representation of the acetylene-based experiment for Pt SACs during XAS monitoring. **b** *Operando* Pt $L_3$ edge EXAFS of different Pt SACs featuring distinct coordination environments as determined by the support and the synthetic procedure (i.e., solvent and thermal activation). **c** MCR analysis of *operando* XAS of $Pt_{SA}$/AC-w-473 (bottom), exposed to acetylene under room temperature (white region, left), temperature ramp (5 K min⁻¹, gray region), and under reaction temperature (473 K, white region, right), together with structural representations of the dynamic

behavior of the Pt SACs (top). **d** Weak interaction quadrant (left) of the 2D HYSCORE EPR spectra of carbonaceous deposits generated over $Pt_{SA}$/AC-w-473 in 12 h on stream, together with anti-diagonal projection of the ¹⁹⁵Pt component (right). The analysis shows that coke forms exclusively over the support yet in proximity of the metal sites, within 5 Å, as depicted by the structural representation in the inset. Reaction conditions: $T_{bed}$ = 473 K, $F_{tot}$ = 15 cm³ min⁻¹, $C_2H_2$:HCl:Ar = 40:44:16, $W_{cat}$ = 0.25 g.

chloride ligands (Pt-Cl CN = 2.4, Supplementary Table S11), the stabilization of $PtCl_2$ species over keto₄ sites is found to be energetically highly favorable (−3.0 eV, with respect to pristine keto₄ sites and an isolated $PtCl_2$ species). This structure, denoted as $PtCl_2$/keto₄, provides an adaptive coordination of the $PtCl_2$ species with oxygen pairs in the carbon (Fig. 7a), which can account for their three-fold coordination with the support.

Upon identification of the active structure, the affinity of the metal atoms for both reactants under reaction conditions is computed. Notably, highly chlorinated Pt species present endergonic adsorption of acetylene (>0.5 eV, Fig. 7a, Supplementary Table S22). Simulation of the reactant competitive adsorption shows that, irrespective of their coordination with the support, $PtCl_2$ species preferentially bind HCl over $C_2H_2$ by at least 0.5 eV, reflecting hydrogen chloride activation as observed in *operando* XAS (*vide supra*, Supplementary Table S23). This is ascribed to the closed coordination shell that highly chlorinated metal atoms exhibit, which prevents the adsorption of acetylene but not of hydrogen chloride as the latter strongly binds the chloride anion. In contrast, the adsorption of acetylene in $PtCl_2$-neighboring C and O sites in the support is found to be exergonic in the range of −0.9 to 0.0 eV (Fig. 7a, Supplementary Table S22), as acetylene can undergo addition reaction to the carbon support forming five-, six-, and seven-membered ring configurations.

Having resolved the nature of the binding sites for each reactant, the reaction profile is computed over the $PtCl_2$/keto₄ structure (Fig. 7b, Supplementary Table S24). Initially, acetylene is adsorbed on metal-neighboring sites in the carbon support (denoted as #), forming the $C_2H_2$# intermediate. Meanwhile, hydrogen chloride is heterolytically

dissociated over the Pt atom (denoted as *) and yields the adsorbed H* and Cl* intermediates, though H* promptly migrates to the support and forms H# (1.19 eV, Fig. 7). This leads to a three-fold coordination with chloride ligands as the resting state for the Pt atoms, in line with the *operando* XAS analysis (Fig. 7b, Supplementary Table S11). Subsequently, H# is transferred from the scaffold to the adsorbed $C_2H_2$# and finally, the formation of VCM is driven by the recovery of the initial metal site coordination via chloride donation. This mechanism presents an activation energy of 0.51 eV (Supplementary Table S24), in reasonable agreement with the experimental value of 0.37 eV (Supplementary Fig. S24). Overall, these results evidence that active structures in single-atom catalysts for acetylene hydrochlorination are ensembles that comprise metal atoms and metal-neighboring sites in the support.

## Discussion

Addressing long-standing questions about their catalytic function, carbons are experimentally proven to work together with metal atoms in fulfilling the acetylene hydrochlorination cycle, as illustrated in Fig. 8. To assess the role of both components, we generated a platform of Pt, Au, and Ru single atoms on commercial activated and non-activated carbons, as well as a nitrogen-doped carbon, exhibiting distinct chemical properties and performance. Furthermore, aiming to shed light on synthesis-structure-performance relations, we investigated the influence of synthetic parameters such as solvent choice and thermal activation temperature, on both the metal and carbon structures. Structural changes of both components were probed from catalyst synthesis to operation, by combining *operando* X-ray absorption

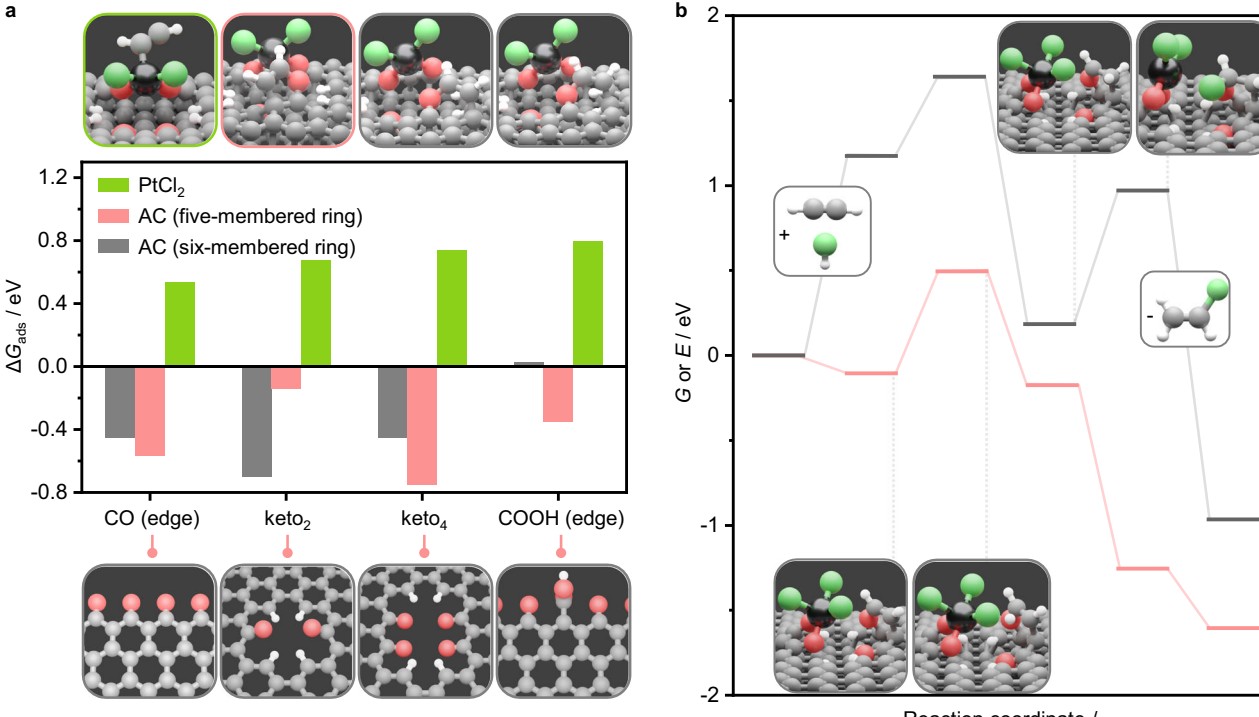

**Fig. 7 | Acetylene-binding site and reaction profile. a** Gibbs free energy of acetylene over $PtCl_2$ and C/O binding sites on distinct cavities, together with structural representations of the different acetylene adsorption configurations (top) and cavity geometries (bottom). Specifically, acetylene adsorbs on the carbon support by forming five- or six-membered rings. In the case of $keto_2$ and $keto_4$ sites, the six-membered configuration rapidly evolves into a seven-membered configuration (top right). **b** Gibbs free energy (gray) and potential energy (red) of the reaction profile over bifunctional metal-carbon sites in the $PtCl_2/keto_4$ active structure.

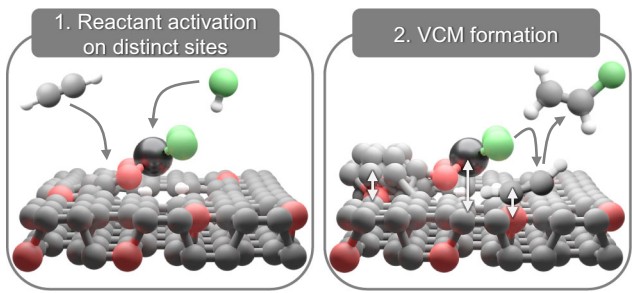

**Fig. 8 | Structure and function of active sites.** Graphical summary of key insights of this work. Active structures comprise metal and carbon sites, respectively activating hydrogen chloride and acetylene (left). Metal atoms mediate chloride supply to acetylene for VCM formation, while carbon surface functionalities regulate acetylene adsorption and coke formation (right).

spectroscopy investigations of metal sites with kinetic studies and electron paramagnetic resonance and X-ray photoelectron spectroscopic analyses of carbonaceous species. Interestingly, the dynamic behavior of metal atoms was found similar across different carbons, despite their dissimilar performance. Likewise, the behavior of metal sites generated by employing the highly oxidizing aqua regia solvent, as opposed to water, remained virtually unchanged, while the resulting carbon modifications led to reduced activity. Metal atoms were found to exclusively activate hydrogen chloride, though the extent of this catalytic function appears to be regulated by their coordination with the support, as determined by surface functionalization and activation temperature in the synthesis (Fig. 8). Conversely, metal-neighboring sites in the support bind acetylene and are responsible for catalyst deactivation by coking (Fig. 8). Specifically, coke formation was ascribed to polymerization of reaction intermediates, yielding carbonaceous

deposits in proximity, *ca*. 5 Å, of the metal sites. These findings evidence that the metal sites and the carbon supports should be selectively designed for optimal hydrogen chloride and acetylene interactions, respectively, to boost the catalytic activity while avoiding fouling by coking. Guided by the resolution of the metal coordination environment under reaction conditions by *operando* XAS, DFT simulations enabled us to propose potential acetylene-activating sites in the carbon and an accessible reaction profile over bifunctional metal-carbon sites. On the basis of these findings, and upon careful equipment design to withstand the reaction corrosiveness, future endeavors employing in situ or *operando* XPS and infrared techniques (e.g., diffuse reflectance infrared Fourier transform spectroscopy) hold promise to offer valuable insights into the nature and dynamic behavior of the carbon functionalities that bind acetylene. Ultimately, these analyses should shed light on the role of carbon supports in catalyzing acetylene hydrochlorination across different metal structures (i.e., single atoms, clusters, and nanoparticles), as nanostructuring strategies can substantially modify the electronic properties and catalytic role of the metal component[40]. Going beyond acetylene hydrochlorination, our results evidence that active structures in single-atom catalysts can comprise metal atoms and neighboring sites in the support. Their co-catalytic function has been recently proposed in other applications[41–44], suggesting that the concept of single-site catalysis should be expanded to ensemble catalysis[45]. As a result, future engineering efforts are encouraged to explore selective design of the metal atom and the surrounding sites in the support to optimize each component's catalytic function at the atomic level and unlock superior performance.

## Methods
### Catalyst preparation
All metal-based catalysts (nominal metal loading 1 wt%) were prepared via an incipient wetness impregnation method, employing the

corresponding metal chlorides as precursors dissolved in deionized water or aqua regia. The obtained solutions were added dropwise to the different carbon supports. Subsequently, all samples were dried at 333 K (heating rate = 5 K min⁻¹, hold time 12 h, static air). The respective nanostructured catalysts were obtained via thermal activation ($T_a$, heating rate = 5 K min⁻¹, hold time 12 h, static air or $N_2$ if $T_a$ > 673 K) and denoted as $M_X$/support-solvent-$T_a$ (M = Pt, Au or Ru; $X$ = single atoms (SA) or nanoparticles (NP), support = NC, AC, or C; solvent = water (w) or aqua regia (a); $T_a$ = 473–1073 K). Further details on the catalyst synthesis and the preparation of the carbon supports are provided in the Supplementary Methods.

## Catalyst characterization

Multiple techniques were employed to characterize the catalytic materials, as summarized in Supplementary Table S2. In particular, the metal dispersion was assessed through X-ray diffraction (XRD) and high-angle annular dark-field scanning transmission electron microscopy (HAADF-STEM). The porous properties of the carbon supports were assessed by Ar sorption at 77 K. The composition and chemical state of the metal atoms and the carbon supports were evaluated by X-ray photoelectron spectroscopy (XPS). The metal oxidation state and coordination environment during synthesis and under reactive environments, as summarized in Supplementary Fig. S26, were monitored by *operando* X-ray absorption spectroscopy (XAS), respectively, by X-ray Absorption Near Edge Spectroscopy (XANES) and extended X-ray absorption fine structure (EXAFS). The interaction of the catalysts with acetylene was studied by temperature-programmed desorption of acetylene coupled to mass spectrometry ($C_2H_2$-TPD-MS). Coke composition and spatial distribution were assessed by electron paramagnetic resonance spectroscopy (EPR). All characterization techniques and procedures are detailed in the Supplementary Methods.

## Catalytic evaluation

The hydrochlorination of acetylene was evaluated at atmospheric pressure in a continuous-flow fixed-bed reactor set-up, as described elsewhere[12]. In a typical test, the catalyst ($W_{cat}$ = 0.25 g) was loaded in the quartz reactor and heated in a He flow to the desired bed temperature ($T_{bed}$ = 473 K). After stabilization for at least 15 min, the reaction mixture (40 vol% $C_2H_2$, 44 vol% HCl, and 16 vol% Ar) was fed at a total volumetric flow of $F_T$ = 15 cm³ min⁻¹, employing a high gas hourly space velocity based on acetylene, $GHSV(C_2H_2)$ = 650 h⁻¹. Reactants and products, including the yield of vinyl chloride, $Y$(VCM), the carbon mass balance, and mass and heat transfer limitations were evaluated according to the protocols described in the Supplementary Methods.

## Computational methods

To gain insights into the interaction of acetylene with distinct metal sites, DFT calculations were performed using the Vienna Ab initio Simulation Package with projector augmented wave core potentials and the PBE-D3 functional, as detailed in the Supplementary Information. In brief, six oxidic defects (keto-, hydroxyl-, epoxide-, and carboxylic-based sites) were populated by Pt single atoms with a varying number of chloride ligands, and evaluated on the basis of stability (i.e., formation energies). Guided by the resolution of the metal coordination environment under reaction conditions provided by *operando* XAS, we focused on di-chlorinated Pt sites stabilized over four keto sites (i.e., a tetra-ketone cavity, keto₄) to compute the reaction profile. All computed structures can be retrieved from the ioChem-BD database[46].

## Data availability

The experimental and computational data presented in the main figures of the manuscript are publicly available through the Zenodo (https://doi.org/10.5281/zenodo.7855567) and ioChem-BD (https://doi. org/10.19061/iochem-bd-1-284) repositories, respectively. Further data supporting the findings of this study are available in the Supplementary Information. All other relevant source data are available from the corresponding author upon request. Source data are provided with this paper.

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

## Acknowledgements

This publication was created as part of NCCR Catalysis (grant number 180544), a National Centre of Competence in Research funded by the Swiss National Science Foundation. The Scientific Center for Optical and Electron Microscopy (ScopeM) at the ETH Zurich and the SuperXAS beamline at PSI, are thanked for access to their facilities. The Spanish Ministry of Science and Innovation is acknowledged for financial support (PID2021-122516OB-I00 and Severo Ochoa Grant MCIN/AEI/10.13039/501100011033CEX2019-000925-S) and the Barcelona Supercomputing Center-MareNostrum (BSC-RES) for providing generous computer resources. A.R.-F. acknowledges funding from the Generalitat de Catalunya and the European Union under Grant 2023 FI-3 00027. The authors are grateful to D. Zindel for preparing gas mixtures, S. Damir for assistance with catalytic testing, Dr. M. Vanni and D. Faust Akl for assistance with XAS measurements, and Dr. S. Mitchell for fruitful discussions.

## Author contributions

J.P.-R. conceived and coordinated all stages of this research. V.G., G.G., and I.S. synthesized the catalysts, contributed to their characterization, and conducted the catalytic tests. A.R.-F. and N.L. conducted the density functional theory simulations. F.K. conducted the electron microscopy analyses. A.H.C. supervised acquisition and evaluation of X-ray absorption spectroscopy data. M.A. and G.J. conducted the electron paramagnetic resonance spectroscopy studies. All authors contributed to the writing of the manuscript.

## Competing interests

The authors declare no competing interests.
