## [Peer Review File · Nature Communications]

REVIEWER COMMENTS

Reviewer #1 (Remarks to the Author):

The paper presents evidence of bifunctionality of carbons and metal atoms in catalyzed acetylene hydrochlorination. It is a topic of interest to the researchers in the related areas. But the paper needs very significant improvement before acceptance for publication. My detailed comments are as follows:

1. In the second paragraph of page 10, the authors suggested that "The acetylene adsorption on the metal sites might be prevented by the presence of chloride ligands". However, in the second graph of page 8, the authors presented that "On the contrary, upon exposure to only acetylene, chloride ligands are removed." Is there any experimental phenomenon indicating that acetylene has bonded on the uncoordinated metal.

2. the authors proposed role of carbon as an "acetylene reservoir." and also presented that "The lack of metal-acetylene interactions in all the examined Pt SACs supported on the diverse

carbons (AC, NC, and C), indicates that the acetylene activation step occurs over the carbon, irrespective of its properties." I wonder what is the identity of the active site that can bind acetylene so strong (even stronger than metal-acetylene interaction) and activate acetylene to undergo VCM formation. What is the mechanism that involved in above process?

Reviewer #2 (Remarks to the Author):

In this manuscript, combining the operando X-ray absorption spectroscopy with other spectroscopic and kinetic analyses, the effect of single metal atom and carbon support in a supported Pt catalyst for the acetylene hydrochlorination has been discussed. The dissociation of HCl owes to the single metal atom, which is in accordance with the work published by Hutchings et al [10.1126/science.aal3439]. On the other hand, C₂H₂ is adsorbed and activated by the activated carbon support, and then the dissociated HCl molecule migrates to activated C₂H₂ to form VCM product, which has not been proposed before. This work can help explain the effect of carbon support in acetylene hydrochlorination, but there are some results that need to be clarified. Before considering accepting the work published in Nature Communications, the authors need to issue the following questions:

Minor questions:

1. There are some distinguishable peaks surrounding C(101) surface in Supplementary Figure S1. Are they related to metals or metallic compounds? What are they?
2. As shown in Supplementary Figure S2, there might be some metal clusters rather than single-atom Pt on the PtSA/NC-a-473 catalyst. The authors need to explain it.
3. On Page 6, line 16: "In line with previous reports, mild activation temperatures (i.e., 473 K) yield single atoms, while harsher thermal treatments (i.e., 673 K) lead to metal sintering on AC and the formation of chloride-free isolated Pt sites on NC that are four-fold coordinated with the support (Figure 2c), as corroborated by extended X-ray absorption fine structure (EXAFS, Supplementary Table S6)." The authors considered the N-doped carbon support but without presenting the coordination results of Pt-N. Please explain it.
4. In Figure 2c, the authors mentioned that Pt species were sintered under the high temperature of 673 K, and the model diagram shows the Pt cluster rather than the single dispersed Pt species in PtSA/AC-w-873 catalyst. According to the caption and model diagram, should PtSA/AC-w-873 be PtNP/AC-w-1073? or 1023? 673? If so, why Pt-Pt peak could not be found in R space? In addition, should PtSA/AC-w-1023 be PtSA/AC-w-1073, please clarify it.
5. In Table S11, why do the fitting parameters of PtSA/AC-w-473-4h catalyst show a significant difference with other catalysts?
6. The ratio of C₂H₂ and HCl reactants in the description of catalytic evaluation is different from that in the caption of Figure 4a. Please clarify it.

Major concerns:

7. In the previous work [10.1021/acssuschemeng.2c07478, 10.1039/D1NJ05120B, 10.1016/j.mcat.2023.113158], C₂H₂-TPD results showed obvious interactions between active metal and C₂H₂ on the Au-, Ru- and Pt-based catalysts, which seem to conflict with this work. The authors need to compare the C₂H₂-TPD results for supported metal catalysts and supports. For example, PtSA/AC-w-473 and AC-w-473.
8. To further explore the effect of support on the activity, contrast experiments need to be performed. Since the catalyst prepared by the aqua regia will cause the oxidation of support, the catalyst with the aqua regia modified support and then the water-impregnated Pt should be prepared. Catalytic evaluation experiments and C₂H₂-TPD characterizations need to be conducted.
9. On page 21, the authors mentioned that "EXAFS analysis evidences two contributions: an unaltered short-bonding one, at $1.85 \pm 0.03 \text{ \AA}$, and a growing long-bonding one, at $2.04 \pm 0.02 \text{ \AA}$ (Supplementary Table S19). Though oxygen and carbon atoms exhibit similar scattering properties in XAS, the short- and long-bonding contributions are in agreement with Pt-C and Pt-O references, respectively (Figure 4b)". Why not use the Pt-C and Pt-O instead of Pt-C/O short and Pt-C/O long, respectively? The same problem is shown in the two types of Pt-Cl coordination with obviously different values of CN and R. Is it caused by the first and second shells?
10. Since the activated carbon support can adsorb C₂H₂, what kind of carbon sites show unique activity? In addition, if the reaction follows the Eley-Rideal mechanism, which means that the gas

C₂H₂ molecule could react with the adsorbed HCl molecule, the Pt-C coordination will not show obvious change during the reaction. In this case, it is also consistent with the results in Figure 5a. How the authors evidence that both C₂H₂ and HCl participate in the reaction with the adsorbed or activated state.

11. The Pt-Cl coordination in the PtSA/NC-w-1073 catalyst did not exist after annealing at 1073 K, is it replaced by Pt-N coordination? Will it affect the HCl dissociation by active metal sites and the catalytic activity for acetylene hydrochlorination?

Reviewer #3 (Remarks to the Author):

In this manuscript, Giulimondi et al. reported bifunctional role of carbon and metal atoms (Pt, Ru and Au) for the acetylene hydrochlorination reaction. The catalysts were characterized by XRD, HAADF-STEM, XPS and XAS spectroscopy and the catalytic tests were carried out in a continuous-flow fixed-bed reactor under relevant conditions. The study is very concrete and the results obtained are very coherent and supported by many evidences such as operando XAS which is challenging and at the same time also risky and I am glad that proper safety assessments were carried out. The EPR (HYSCORE) study is also interesting as coking is a huge challenge for this industrially relevant reaction. I am sure that in the near future this process will be conducted based on the transition metals instead of hazardous supported HgCl₂ as catalyst. The manuscript is written very nicely and the presentation of figures and graphics are very clear.

The manuscript can be further improved for general readers by implementing the following points.

1. Since the manuscript shows lots of results based on XAS, it will be helpful for the readers if the authors also include the spectra of PtO₂, PtCl₂ and Pt metal in the figure 2d as they did in the figure S4.

2. The table related to the EXAFS fitting parameters missed two important things (a) R-factor and (b) Amplitude reduction factor. Is there any reason that the authors did not include these?

3. EXAFS spectra are usually recorded at room temperature to have minimum thermal disorder which influences the Debye Waller factor. It seems that majority of the spectra are collected at 473 K and I am bit surprised that the Debye-Waller parameters are still showed relatively lower values. In the fitting, what kind of model were taken into account?

4. I am little bit worried about the assignment of Pt-Pt shell in figure 2c. The Pt-Pt shell that is assigned at 2.2 Å seems to be shorter (even without phase correction) than usual as the authors correctly showed in table S15. May be the authors also compare the R space of Pt foil in the same figures to make it clear.

5. In page 6, in the sentence 11, instead of “gradually resembling the PtO₂ reference”, I think the features of H₂PtCl₆ resembles the most of the features. Hence, the authors should consider changing it.

6. It is not clear what is step v in the page 9 refers to (sentence 10, “results, analysis of the Cl 2p XPS spectra after exposure to acetylene only (step v) reveals a”

7. Line 14, page 9, it is mentioned that “Furthermore, analysis of the O 1s spectra shows that the oxygen functionalities undergo a slight reduction, pointing toward their participation in the reductive formation of VCM”. Is it under the reaction conditions?

8. It will be very informative to the general reader to know the parameters of MCR analysis. The authors can consider including this information in the ESI.

9. For the future scope (beyond this work), studies like FTIR (DRIFT) might be very informative even though I can understand that to design cell will be extremely challenging under such corroding environment. However, to observe interactions between M-acetylene might be useful.

Additional questions

What are the noteworthy results? The results presented here are very insightful for such an industrially relevant process

Will the work be of significance to the field and related fields? How does it compare to the established literature? Very significant

Does the work support the conclusions and claims, or is additional evidence needed? Yes, very coherent study with evidences

Are there any flaws in the data analysis, interpretation and conclusions? Do these prohibit publication or require revision?

Is the methodology sound? Does the work meet the expected standards in your field? Yes, the methodology applied is commendable

Is there enough detail provided in the methods for the work to be reproduced? Yes

Manuscript NCOMMS-23-20207 - Response to Reviewers

Comments in *blue* - Replies in black - Actions in **bold**

Important note: Indicated page, line, or figure numbers refer to the revised manuscript and/or Supplementary Information with changes highlighted.

Reviewer #1

The paper presents evidence of bifunctionality of carbons and metal atoms in catalyzed acetylene hydrochlorination. It is a topic of interest to the researchers in the related areas. But the paper needs very significant improvement before acceptance for publication.

We thank the Reviewer for recognizing the significance of our study, relevant to the fields of acetylene hydrochlorination and single-atom catalysis. Their thoughtful comments, addressed below, prompted us to conduct additional kinetic tests and experimentally-guided density functional theory (DFT) simulations to further elaborate on the active site structure and the function of each component partaking in the catalytic cycle.

My detailed comments are as follows:

1. In the second paragraph of page 10, the authors suggested that "The acetylene adsorption on the metal sites might be prevented by the presence of chloride ligands". However, in the second graph of page 8, the authors presented that "On the contrary, upon exposure to only acetylene, chloride ligands are removed." Is there any experimental phenomenon indicating that acetylene has bonded on the uncoordinated metal.

This is a central point, which relates to comment #10 of Reviewer #2. The *operando* XAS analysis presented in our contribution does not evidence any metal-acetylene interactions under reaction conditions, as discussed in detail for the Pt_{SA}/AC-w-473 catalyst (page 8, lines 24-26; page 9, lines 1-3 and 16-22). **This has been corroborated by extensive computational studies (page 33, Figure 7a; page 16, lines 8-19).** DFT results evidence that highly chlorinated metal atoms present endergonic adsorption (Gibbs energies of acetylene > 0.5 eV, page S37, Supplementary Table S22). This is exemplified by di-chlorinated Pt atoms, the most likely metal speciation in the optimal working catalyst (page S37, Supplementary Table S22).

To further validate the lack of metal-acetylene interaction, we exposed as-prepared Pt SACs to C₂H₂ only from room temperature, to maximize adsorption, to reaction temperature (*i.e.*, 473 K). *Operando* XAS analysis shows that metal dechlorination occurs only upon increasing the temperature, suggesting that the dechlorination process is linked to VCM formation. More specifically, EXAFS analysis of the as-prepared and C₂H₂-exposed catalysts shows only a partial loss in the Pt-Cl contribution (coordination number, CN = 3.1 and 0.7, respectively). This is counterbalanced by a stronger long-bonding Pt-C/O contribution (CN = 0.2 and 2.0, respectively), while the short-bonding Pt-C/O one remains unaltered (CN = 0.9 and 1.0, respectively). Indeed, the total coordination number of the metal atoms determined by *operando* XAS, encompassing Pt-Cl and Pt-C/O contributions, remains *ca.* 4 prior and after exposure to C₂H₂ (page S34, Supplementary Table S19). The attribution of the increased Pt-C/O interaction to metal-acetylene interactions cannot be ruled out, **as it has been clarified in the manuscript (page 13, lines 9-10).** However, the long-bonding Pt-C/O contribution is in agreement with the Pt-O references rather than Pt-C ones (*J. Phys. Chem. B* **104**, 1998, (2000); *Catal. Lett.* **8**, 283, (1991)). This suggests that the partial removal of chloride ligands leads the uncoordinated metal atom to re-coordinate to the support rather than adsorbing C₂H₂ as a ligand. In line with this, DFT simulations predict an increase in the metal-support binding energy of at least 1.2 eV upon partial metal dechlorination (*i.e.*, average loss of two chloride ligands; page S36, Supplementary Table S21), irrespective of the metal site structure. **These considerations relating to both experimental and computational investigations are now discussed in the manuscript and Supplementary Information (page 33, Figure 7a; page 13, lines 9-10; page 16, lines 8-19; pages S36-S38, Supplementary Tables S21-S23).**

2. The authors proposed role of carbon as an "acetylene reservoir." and also presented that "The lack of metal-acetylene interactions in all the examined Pt SACs supported on the diverse carbons (AC, NC, and C), indicates that the acetylene activation step occurs over the carbon, irrespective of its properties." I wonder what is the identity of the active site that can bind acetylene so strong (even stronger than metal-acetylene interaction) and activate acetylene to undergo VCM formation. What is the mechanism that involved in above process?

We thank the Reviewer for the insightful question, which also relate to comment #10 of Reviewer #2. We clarify that the presented spectroscopic and kinetic investigations point to the carbon supports as responsible for adsorbing and activating C_2H_2 , but no conclusion could be drawn on the nature of the sites, requiring further investigations, **as stated in the manuscript (page 15, lines 1-5)**. Indeed, carbon-based materials present different functionalities that can bind C_2H_2 (*Nature* **436**, 238 (2005); *Chem. Mater.* **31**, 4919 (2019)).

To gain further insights into the active site structure and the function of each component in fulfilling the catalytic cycle, **we have conducted additional kinetic and computational investigations**. Efforts were focused on the $Pt_{SA}/AC-w-473$ catalyst, selected for its high activity and unparalleled stability (*Nat. Catal.* **3**, 376 (2020)). Derivation of partial reaction orders corroborated the adsorption of both reactants over the catalyst surface (please see our reply to comment #10 of Reviewer #2, page 15, lines 6-12; page S66, Supplementary Figure S23). To identify the reactant binding sites, DFT simulations were performed on active structures featuring chlorinated Pt atoms anchored over diverse potential O-functionalities in AC. *Operando* XAS analysis indicates coordination of Pt atoms with at least two chloride ligands, which preferentially bind HCl over C_2H_2 , as detailed in our reply to comment #1. Conversely, C_2H_2 adsorption on the chlorinated Pt atoms is endergonic ($> ca. 0.5$ eV; page S37, Supplementary Table S22). Thereafter, the adsorption of C_2H_2 was computed on different functionalities in the support that neighbor the di-chlorinated Pt species, resulting in Gibbs energy values ranging from -0.9 to 0.0 eV. This exergonic character can be assigned to the ability of C_2H_2 to undergo addition reactions with the carbon support by forming five- and six-membered rings (page 16, lines 16-19; page 33, Figure 7a). Finally, guided by the *operando* XAS and kinetic analyses, a reaction profile was proposed and found to be energetically viable, as detailed in our reply to comment #10 of Reviewer #2.

The new kinetic and computational results have been integrated and discussed in the manuscript and the Supplementary Information (page 33, Figure 7; page 15, lines 1-26; page 16, lines 1-26; page 17, lines 1-7; pages S36-S39, Supplementary Tables S21-S24).

Reviewer #2

In this manuscript, combining the operando X-ray absorption spectroscopy with other spectroscopic and kinetic analyses, the effect of single metal atom and carbon support in a supported Pt catalyst for the acetylene hydrochlorination has been discussed. The dissociation of HCl owes to the single metal atom, which is in accordance with the work published by Hutchings et al [10.1126/science.aal3439]. On the other hand, C₂H₂ is adsorbed and activated by the activated carbon support, and then the dissociated HCl molecule migrates to activated C₂H₂ to form VCM product, which has not been proposed before. This work can help explain the effect of carbon support in acetylene hydrochlorination, but there are some results that need to be clarified.

We thank the Reviewer for appreciating the novelty of our findings regarding the uncovering of the co-catalytic role of carbon supports in metal-catalyzed acetylene hydrochlorination. By addressing their constructive criticism, we were able to further strengthen the quality and impact of our study, as well as the clarity of the messages. Specifically, **additional kinetic and experimentally-guided computational investigations, as well as C₂H₂ adsorption analyses, have been conducted** to further support our conclusions and gain deeper insights into the active site structure and functions in fulfilling the catalytic cycle. Each point is addressed below with a description of the actions taken upon revision.

Before considering accepting the work published in Nature Communications, the authors need to issue the following questions:

Minor questions:

1. There are some distinguishable peaks surrounding C(101) surface in Supplementary Figure S1. Are they related to metals or metallic compounds? What are they?

The diffraction peaks in question are attributed to incomplete activation of the commercial AC support employed in the study (*Sci. Rep.* **10**, 2563 (2020)), **as detailed in the Supplementary Information (page S41, lines 5-6)**. In line with this, the diffraction peaks are not observed in NC-based catalysts.

2. As shown in Supplementary Figure S2, there might be some metal clusters rather than single-atom Pt on the Pt_{SA}/NC-a-473 catalyst. The authors need to explain it.

We thank the Reviewer for the comment. We have newly conducted the investigation by microscopy of the Pt_{SA}/NC-a-473 catalyst. The analysis corroborates the atomic dispersion of the metal, as visualized in the micrograph that **has been included in the Supplementary Information (page S43, Supplementary Figure S2)**.

3. On Page 6, line 16: "In line with previous reports, mild activation temperatures (i.e., 473 K) yield single atoms, while harsher thermal treatments (i.e., 673 K) lead to metal sintering on AC and the formation of chloride-free isolated Pt sites on NC that are four-fold coordinated with the support (Figure 2c), as corroborated by extended X-ray absorption fine structure (EXAFS, Supplementary Table S6)." The authors considered the N-doped carbon support but without presenting the coordination results of Pt-N. Please explain it.

This is a relevant observation, which relates to comment #11. We realize N was not included in Supplementary Table S6 among the light-scattering atoms, which **has now been amended (page S19, Supplementary Table S6)**. Indeed, distinguishing between N, C, and O is highly challenging in EXAFS analysis, owing to their similar scattering properties (*Chem. Rev.* **121**, 882 (2021)). Still, the presented operando XAS investigation indicates a four-fold coordination of the Pt atoms with the support, identifying a total metal coordination number with light-scattering atoms of 4.3.

4. In Figure 2c, the authors mentioned that Pt species were sintered under the high temperature of 673 K, and the model diagram shows the Pt cluster rather than the single dispersed Pt species in Pt_{SA}/AC-w-873 catalyst. According to the caption and model diagram, should Pt_{SA}/AC-w-873 be Pt_{NP}/AC-w-1073? or 1023? 673? If so, why Pt-Pt peak could not be found in R space? In addition, should Pt_{SA}/AC-w-1023 be Pt_{SA}/AC-w-1073, please clarify it.

We thank the Reviewer for pointing out the typo in the sample code in Figure 2c, which should denote “Pt_{NP}/AC-w-873” instead of “Pt_{SA}/AC-w-873”, **as now corrected (page 25, Figure 2c). As clarified in the revised manuscript (page 6, lines 21-23)**, the formation of clusters is observed under temperatures above 673 K (*Nat. Catal.* **3**, 376 (2020)), which is attributable to the partial loss in chloride ligands that stabilize the Pt species as single atoms. In line with this, our *operando* XAS investigation of AC-supported Pt species undergoing thermal treatment at 873 K, as indicated by the sample code, evidences the formation of metallic clusters reflected in a Pt-Pt CN of 1.3 and a low Pt-Cl CN of 0.4 (page S19, Supplementary Table S6). Furthermore, the HAADF-STEM investigation of Pt_{NP}/AC-w-873 shows mixed metal speciation encompassing nanoparticles and low-nuclearity species (page S43, Supplementary Figure S2), indicating the uncontrolled nature of the sintering process, **as highlighted in the Supplementary Information (page S43, Supplementary Figure S2, lines 3-5).**

5. In Table S11, why do the fitting parameters of Pt_{SA}/AC-w-473-4h catalyst show a significant difference with other catalysts?

We thank the Reviewer for pointing out this inconsistency. We deeply regret that the different fitting parameters for the Pt_{SA}/AC-w-473-4h catalyst were misreported, and **they have been corrected (page S24, Supplementary Table S11).**

6. The ratio of C₂H₂ and HCl reactants in the description of catalytic evaluation is different from that in the caption of Figure 4a. Please clarify it.

We thank the Reviewer for noting the inconsistency between the caption of Figure 4a and the description of the catalytic evaluation description in the Methods section (page 20, line 5). The reporting of the C₂H₂ and HCl volume percentages in the caption of Figure 4a was reversed and **has been corrected (page 28, line 5).**

Major concerns:

7. In the previous work [[10.1021/acssuschemeng.2c07478](https://doi.org/10.1021/acssuschemeng.2c07478), [10.1039/D1NJ05120B](https://doi.org/10.1039/D1NJ05120B), [10.1016/j.mcat.2023.113158](https://doi.org/10.1016/j.mcat.2023.113158)], C₂H₂-TPD results showed obvious interactions between active metal and C₂H₂ on the Au-, Ru- and Pt-based catalysts, which seem to conflict with this work. The authors need to compare the C₂H₂-TPD results for supported metal catalysts and supports. For example, Pt_{SA}/AC-w-473 and AC-w-473.

The conflicting results likely arise from the *ex situ* versus *operando* conditions of the C₂H₂-TPD and XAS analyses, respectively. C₂H₂-TPD is a valuable approach to probing the interaction of a reactant with the catalyst surface ‘off-line’. However, the technique cannot resolve either the nature of the adsorption site or the catalyst dynamic behavior under reaction conditions, as surface saturation by C₂H₂ is carried out (i) at room temperature and (ii) in the absence of HCl, thus excluding reactant competitive adsorption. Therefore, species such as chloride-free metal atoms, as in the case of the mentioned Ru SACs (*Mol. Catal.* **543**, 113158 (2023)), might bind C₂H₂ in the TPD analysis but might show greater affinity for HCl under reaction conditions and prevent the competitive adsorption of C₂H₂. Additionally, the presence of metal clusters and nanoparticles, as respectively evidenced by the EXAFS and microscopy analyses of the mentioned Au and Pt catalysts (*New J. Chem.*, **46**, 3738 (2022); *ACS Sustainable Chem. Eng.* **11**, 3103 (2023)), further complicates the analysis of acetylene-metal atom interactions. As a result, the different metal speciation hinders a direct comparison with our work. We highlight that our contribution focuses on understanding the catalytic roles of single metal atoms and

carbon supports in catalyzing acetylene hydrochlorination, and does not encompass carbon-supported metal clusters or nanoparticles for which no conclusions are drawn as they would require dedicated studies, **as clarified in the amended manuscript (page 18, lines 14-17).**

Despite these limitations, we agree with the Reviewer that the comparison of C₂H₂-TPD profiles of the supported chlorinated metal single atoms and the bare support can offer insights into the C₂H₂ binding sites. Accordingly, **we have conducted C₂H₂-TPD analysis of the bare AC-w-473**, whose profile shows comparable adsorption properties as those of Pt_{SA}/AC-w-473. Though not fully reflective of acetylene adsorption over the catalyst surface under reaction conditions, these results suggest that the integration of chlorinated Pt atoms does not affect the ability of the carbon support to bind acetylene reinforcing the role of the carbon in the adsorption process. **The additional C₂H₂-TPD analysis and related discussion have been included in the manuscript and Supplementary Information (page 8, lines 12-15; page S50, Supplementary Figure S8).**

8. To further explore the effect of support on the activity, contrast experiments need to be performed. Since the catalyst prepared by the aqua regia will cause the oxidation of support, the catalyst with the aqua regia modified support and then the water-impregnated Pt should be prepared. Catalytic evaluation experiments and C₂H₂-TPD characterizations need to be conducted.

We thank the Reviewer for the valuable suggestion. **We have catalytically evaluated the performance of the suggested catalyst**, denoted as Pt_{SA}/(AC-a-473)-w-473, **and conducted C₂H₂-TPD analysis.** The catalyst presents comparable catalytic activity and C₂H₂ adsorption properties to those of the aqua regia-derived Pt SAC, Pt_{SA}/AC-a-473. These results suggest that the impregnation of the AC support with aqua regia alters the carbon support properties, as both Pt_{SA}/(AC-a-473)-w-473 and Pt_{SA}/AC-a-473 catalysts present reduced catalytic activity (ca. -30% in VCM yield) and C₂H₂ adsorption capacity compared with the water-derived analog, Pt_{SA}/AC-w-473. These findings, now **discussed and presented in the manuscript and Supplementary Information (page 8, lines 6-12, page S50, Supplementary Figure S8)**, corroborate the key role of the carbon support in fulfilling the acetylene hydrochlorination cycle.

9. On page 21, the authors mentioned that “EXAFS analysis evidences two contributions: an unaltered short-bonding one, at $1.85 \pm 0.03 \text{ \AA}$, and a growing long-bonding one, at $2.04 \pm 0.02 \text{ \AA}$ (Supplementary Table S19). Though oxygen and carbon atoms exhibit similar scattering properties in XAS, the short- and long-bonding contributions are in agreement with Pt-C and Pt-O references, respectively (Figure 4b)”. Why not use the Pt-C and Pt-O instead of Pt-C/O short and Pt-C/O long, respectively? The same problem is shown in the two types of Pt-Cl coordination with obviously different values of CN and R. Is it caused by the first and second shells?

We appreciate the Reviewer's observations. Still, despite the Pt-C/O_{short} and Pt-C/O_{long} contributions respectively agree with Pt-C and Pt-O references, the similar scattering properties of C and O atoms (*Chem. Rev.* **121**, 882 (2021)) do not enable us to unambiguously discriminate between Pt-C and Pt-O bonds. As a result, we believe that denoting the Pt-C/O_{short} and Pt-C/O_{long} contributions as Pt-C and Pt-O, respectively, might be misleading as it would suggest unequivocal distinction between Pt-neighboring C and O atoms, which is not possible in EXAFS. Conversely, we agree with the Reviewer on the unclear denotation of the two Pt-Cl contributions at different *R* values, which correspond to the first and second coordination shells, as remarked by them. Therefore, the Pt-Cl contribution relating to the second coordination shell **has been indicated as such in all relevant tables in the Supplementary Information (page S19, Supplementary Table S6; pages S24-S26, Supplementary Tables S11-S12; pages S29-S35, Supplementary Tables S15-S20).**

10. Since the activated carbon support can adsorb C_2H_2 , what kind of carbon sites show unique activity? In addition, if the reaction follows the Eley-Rideal mechanism, which means that the gas C_2H_2 molecule could react with the adsorbed HCl molecule, the Pt-C coordination will not show obvious change during the reaction. In this case, it is also consistent with the results in Figure 5a. How the authors evidence that both C_2H_2 and HCl participate in the reaction with the adsorbed or activated state.

We appreciate the Reviewer's insightful questions, which relate to comment #2 of Reviewer #1. We agree with the Reviewer that, while the presented *operando* XAS analysis indicates HCl activation occurring over the metal atoms (page 8, lines 24-26), it does not exclude the reaction of C_2H_2 molecules in the gas-phase with the activated HCl on the metal sites. Nevertheless, this possibility does not agree with complementary experimental investigations. First, C_2H_2 adsorption over the carbon support is evidenced by C_2H_2 -TPD analyses (please see our reply to comment #7; page 8, lines 6-15; page S50, Supplementary Figure S8). Second, the participation of C_2H_2 in its adsorbed state on the support is substantiated by (i) coke formation over metal-neighboring sites in the support as the leading deactivation pathway, as indicated by EPR and *operando* XAS analyses (page 32, Figure 6d; page 8, lines 20-24; page 14, lines 1-24), and (ii) the reduced activity of the aqua regia-derived SACs compared to the water-derived analogs despite the similar metal behavior under reactive environments monitored by *operando* XAS (page 26, Figure 3a; page 7, lines 23-26; page 8, lines 1-6; page 28, Figure 4e; page 30, Figure 5a; page 12, lines 1-7). This experimental evidence is in high agreement with the key role of carbon in generating performing metal-based catalysts compared to any other support material (*Nat. Commun.* **12**, 4016 (2021); *Chem. Commun.* **53**, 11733 (2017); *Green Chem.* **20**, 2412 (2018)).

Still, to further corroborate the participation of both reactants in their adsorbed state in the reaction, **additional kinetic and experimentally-guided DFT investigations have been conducted**. These were focused on $Pt_{SA}/AC-w-473$, selected for its high activity and unparalleled stability (*Nat. Catal.* **3**, 376 (2020)). First, the C_2H_2 and HCl partial reaction orders were derived. The obtained values (0.60 and 0.48, respectively), lower than 1, suggest the adsorption of both reactants over the catalyst surface for the fulfillment of the catalytic cycle (*J. Catal.* **214**, 130 (2003)). Thereafter, the bifunctional role of carbon and metal sites was theoretically investigated over a di-chlorinated Pt atom stabilized on four-fold keto cavity, denoted as $PtCl_2/keto_4$, in line with the metal site structure identified by *operando* XAS (page S24, Supplementary Table S11). DFT simulations reaffirmed that the metal site in the $PtCl_2/keto_4$ structure activates HCl, while adsorption of C_2H_2 on a metal-neighboring keto group was found energetically favorable. Finally, a plausible reaction profile over the bifunctional metal-carbon active structure was identified, and supported by the reasonable agreement between the computed activation energy and the experimentally-derived value (0.51 vs 0.37 eV, respectively).

The new kinetic and computational results have been presented and discussed in the manuscript and Supplementary Information (page 33, Figure 7; page 15, lines 1-26; page 16, lines 1-26; page 17, lines 1-7; pages S36-S39, Supplementary Tables S21-S24).

11. The Pt-Cl coordination in the $Pt_{SA}/NC-w-1073$ catalyst did not exist after annealing at 1073 K, is it replaced by Pt-N coordination? Will it affect the HCl dissociation by active metal sites and the catalytic activity for acetylene hydrochlorination?

The Reviewer is right, Pt-Cl bonds are replaced by Pt-N ones upon thermal activation at 1073 K. Under harsh thermal conditions, the chloride ligands deriving from the H_2PtCl_6 precursor are fully removed. This yields four-fold coordinated Pt atoms with the support that exhibit a square-planar configuration, as corroborated by the *operando* XAS analysis herein presented (page S34, Supplementary Table S19) and by a recent computational investigation simulating the synthetic process (*Adv. Funct. Mater.* **32**, 2206513 (2022)). The high stability of the square-planar configuration of the Pt atoms results in reduced activity that is comparable to the one of the bare NC support, **as shown in the updated Figure 5c (page 30)**. This is assigned to the reduced ability of the Pt atoms to activate HCl (*Nat. Catal.* **3**, 376 (2020)).

Reviewer #3

In this manuscript, Giulimondi et al. reported bifunctional role of carbon and metal atoms (Pt, Ru and Au) for the acetylene hydrochlorination reaction. The catalysts were characterized by XRD, HAADF-STEM, XPS and XAS spectroscopy and the catalytic tests were carried out in a continuous-flow fixed-bed reactor under relevant conditions. The study is very concrete and the results obtained are very coherent and supported by many evidences such as operando XAS which is challenging and at the same time also risky and I am glad that proper safety assessments were carried out. The EPR (HYSCORE) study is also interesting as coking is a huge challenge for this industrially relevant reaction. I am sure that in the near future this process will be conducted based on the transition metals instead of hazardous supported HgCl₂ as catalyst. The manuscript is written very nicely and the presentation of figures and graphics are very clear.

We warmly appreciate the positive feedback of Reviewer #3 and thank them for recognizing the breadth of our experimental efforts, despite the challenging corrosive nature of acetylene hydrochlorination, and the coherence of the insights provided by the diverse analyses conducted. By addressing their constructive comments, we were able to improve the quality and clarity of our contribution. All inquiries are addressed below point-by-point with the corresponding actions taken.

The manuscript can be further improved for general readers by implementing the following points.

1. Since the manuscript shows lots of results based on XAS, it will be helpful for the readers if the authors also include the spectra of PtO₂, PtCl₂ and Pt metal in the figure 2d as they did in the figure S4.

We fully agree with the Reviewer on the importance of comparing the MCR spectral components shown in Figure 2d with reference spectra. The MCR analysis **has been refined (page 25, Figure 2d; page 7, line 1-6)**, yielding three spectral components that are now directly compared with the H₂PtCl₆, PtCl₂, and PtO₂ references. These were selected as they present white line features similar to those of the MCR components, as discussed in our reply to comment #5. To avoid cluttering Figure 2d, **the spectral components, and related references, have been included in the Supplementary Information (pages S47-S48, Supplementary Figures S5-S6)**. Further elaboration on the refined MCR analysis is provided in our reply to comment #5.

2. The table related to the EXAFS fitting parameters missed two important things (a) R-factor and (b) Amplitude reduction factor. Is there any reason that the authors did not include these?

The tables in the Supplementary Information have been amended to include both the R-factor and the amplitude reduction factor (page S19, Supplementary Table S6; pages S24-S35, Supplementary Tables S11-S20). The R-factors demonstrate high agreement between the fitting models employed and the experimental data, as explicitly shown for both the *R*- and *k*-spaces in the Supplementary Information (pages S52-S53, Supplementary Figures S9-S10; pages S55-S61, Supplementary Figures S12-S18). The amplitude reduction factors were refined from fitting to the corresponding metal foils (yielding 0.79, 0.85, and 0.75 for Pt, Au, and Ru, respectively), **as detailed in the Supplementary Information (page S5, lines 23-24)**.

3. EXAFS spectra are usually recorded at room temperature to have minimum thermal disorder which influences the Debye Waller factor. It seems that majority of the spectra are collected at 473 K and I am bit surprised that the Debye-Waller parameters are still showed relatively lower values. In the fitting, what kind of model were taken into account?

We appreciate the Reviewer's observation, which prompted us to refine the fitting of the EXAFS spectra collected at 473 K. As a result, the fitting of all spectra collected at has been updated employing refined Debye-Waller factors. Specifically, we note a slight increase in the Pt-Cl Debye-Waller factor, which in turn results in a slight increase Pt-Cl coordination numbers at 473 K. Upon refining all fitting parameters carefully, **all relevant tables and figures have been corrected accordingly (page 28, Figure 4;**

page 30, Figure 5; pages S24-S33, Supplementary Tables S11-S18; pages S52-S53, Supplementary Figure S9-S10; pages S55-S60, Supplementary Figures S12-S17).

4. I am little bit worried about the assignment of Pt-Pt shell in figure 2c. The Pt-Pt shell that is assigned at 2.2 Å seems to be shorter (even without phase correction) than usual as the authors correctly showed in table S15. May be the authors also compare the R space of Pt foil in the same figures to make it clear.

We thank the Reviewer for pointing out the misassignment of the Pt-Pt contribution in Figure 2c. **We have included the EXAFS spectrum of the Pt foil**, together with those of H₂PtCl₆, PtCl₂, and PtO₂ references, **in the Supplementary Information (page S46, Supplementary Figure S4)**, showing that the Pt-Pt shell manifests at ca. 2.45 Å in the R-space. **The assignment of the Pt-Pt contribution in Figure 2c has been corrected (page 25, Figure 2c).**

5. In page 6, in the sentence 11, instead of “gradually resembling the PtO₂ reference”, I think the features of H₂PtCl₆ resembles the most of the features. Hence, the authors should consider changing it.

To better explore the nature of the dynamic changes, **we have conducted a singular value decomposition analysis of Pt L₃ edge XAS spectra of H₂PtCl₆ impregnated on both AC and NC during thermal activation**, showing the three components are needed to minimize the eigenvalue when conducting the MCR analysis of the process (**page 7, lines 1-2, page S47, Supplementary Figure S5**). As a result, **the MCR analysis has been refined (page 25, Figure 2d)**, integrating three components. These are individually compared with the H₂PtCl₆, PtCl₂, and PtO₂ references, respectively reflecting the evolving “chlorinated”, “partially dechlorinated”, and “support-anchored” metal speciation during the thermal treatment (**page S48, Supplementary Figure S6**). Prominent dechlorination is observed at first, mirrored by a decreasing contribution of the “chlorinated” component in favor of the “partially dechlorinated” one (page 25, Figure 2d; page S48, Supplementary Figure S6). This is followed by a shift in the white line position toward higher energy, as reflected by the “support-anchored” spectral component whose white line position aligns with the one of the PtO₂ reference, though the latter presents a greater intensity. **The revised MCR analysis and the related discussion have been included in the manuscript and Supplementary Information (page 7, lines 1-6; page 25, Figure 2d; pages S47-S48, Supplementary Figures S5-S6).**

6. It is not clear what is step v in the page 9 refers to (sentence 10, “results, analysis of the Cl 2p XPS spectra after exposure to acetylene only (step v) reveals a”

We thank the Reviewer for pointing out the unclear sentence. The sample therein discussed was exposed to a sequence of six reactive environments (page 28, Figure 4c), to probe the effect of individual reactants and the reaction mixture on the catalyst structure (page 9, lines 8-15). The sentence in question describes the alterations in the catalyst chlorination degree after exposure to acetylene only, corresponding to the fifth reaction environment of the aforementioned sequence. **This has been clarified in the manuscript (page 10, lines 5-6).**

7. Line 14, page 9, it is mentioned that “Furthermore, analysis of the O 1s spectra shows that the oxygen functionalities undergo a slight reduction, pointing toward their participation in the reductive formation of VCM”. Is it under the reaction conditions?

The XPS analyses presented in the study were conducted on catalysts that were exposed to the relevant reactive environments, as indicated by the sample code, in the laboratory set-up (page S6, lines 1-3). **This has been clarified in the manuscript and Supplementary Information (page 10, line 3; page S4, lines 2-4).** To date, operating under ambient pressure and corrosive environments poses major practical challenges to reliably and safely conducting *in situ* or *operando* XPS analysis, requiring synchrotron beam, at a soft X-ray line, and *ad hoc* equipment design (*ACS Catal.* **11**, 1464 (2021); *ACS Catal.* **2**, 2269 (2012)). As a result, conducting *in situ* or *operando* XPS experiments in the short- and

mid-term is unfeasible. Nevertheless, such an endeavor would be highly valuable for resolving the chemical state of both carbon functionalities and metal species, and should be pursued in future studies, **as discussed in the manuscript (page 18, lines 10-14).**

8. It will be very informative to the general reader to know the parameters of MCR analysis. The authors can consider including this information in the ESI.

The description of the MCR analysis method has been included in the Supplementary Information (page S6, lines 7-11).

9. For the future scope (beyond this work), studies like FTIR (DRIFT) might be very informative even though I can understand that to design cell will be extremely challenging under such corroding environment. However, to observe interactions between M-acetylene might be useful.

Operando infrared spectroscopy methods could, indeed, probe the nature and dynamic behavior of the acetylene binding sites in the catalyst. Not only could they enable distinction between metal and support sites but also resolution of the carbon functionalities. Nevertheless, as pointed out by the Reviewer, the highly corrosive nature of the reaction requires *ad hoc* cell design. This is further complicated by the reduced signal-to-noise ratio that carbons exhibit due to their opaque nature (*Carbon* **26**, 889 (1988)). Therefore, we agree with the Reviewer that adequate execution of these experiments will need dedicated time and effort in a future investigation. As this undertaking goes beyond the scope of the herein-presented first study on the role of carbon and metal atoms in catalyzing acetylene hydrochlorination, the potential insights that future studies employing *operando* infrared spectroscopy methods could offer **have been discussed in the manuscript (page 18, lines 10-14).**

REVIEWERS' COMMENTS

Reviewer #2 (Remarks to the Author):

I appreciate the efforts made by the authors in addressing the concerns. The authors have relatively adequately addressed the raised issues. Therefore, I would like to recommend its acceptance for publication.

Reviewer #3 (Remarks to the Author):

The authors have conducted great revision including some new insightful DFT work and raised all my concerns. Hence I do not have any further comments or suggestions and I recommend the manuscript for publication.

Manuscript NCOMMS-23-20207A - Response to Reviewers

Comments in *blue* - Replies in black

Reviewer #2

I appreciate the efforts made by the authors in addressing the concerns. The authors have relatively adequately addressed the raised issues. Therefore, I would like to recommend its acceptance for publication.

We thank Reviewer #2 for appreciating our efforts to address the Reviewers' comments and strengthen the quality of our contribution.

Reviewer #3

The authors have conducted great revision including some new insightful DFT work and raised all my concerns. Hence I do not have any further comments or suggestions and I recommend the manuscript for publication.

We are delighted to read Reviewer #3's positive feedback on our revised contribution and the computational insights therein integrated, consolidating the impact of our work.